# TOWARD DOMAIN TRANSLATION WITH MONOLINGUAL DOMAIN DATA ONLY

## ABSTRACT

Neural machine translation (NMT) is very sensitive to domain shifts requiring a carefully designed fine-tuning strategy to avoid catastrophic forgetting problems when adapting to a new domain. Fine-tuning usually relies on high quality in-domain data, but constructing a sufficient amount of parallel data for training poses challenges even for fine-tuning. In contrast, domain-specific monolingual resources are more accessible when compared with bilingual data. Therefore, we challenge the domain adaptation of a general NMT model using only features obtained from a small amount of monolingual data. We regard the task as an instance of domain shifts, and adopt energy-based models (EBMs) and approximate these EBMs using Conditional Distributional Policy Gradients (CDPG). Recent work has applied CDPG with a small number of EBMs for NMT models limiting the capacity for domain shifts, but we construct a large number of EBMs considering the entire domain-specific data, i.e., unigram distribution, and perform fine-tuning according to their constraints. Our results show that fine-tuning using a large number of EBMs can achieve a robust domain shift without causing catastrophic forgetting, demonstrating a robust domain shift using only a small amount of monolingual resources.

## 1 INTRODUCTION

Thanks to the development of crawling technology and the construction of corpora (Tiedemann, 2012; Bañón et al., 2020; Morishita et al., 2022), we have access to abundant parallel translation data, resulting in the development of high-performance pre-trained NMT models. However, it has been pointed out that NMT models suffer from performance degradation when translating text from the domains different from the domain of the training corpus due to the mismatch of the domain-specific terminologies (Koehn & Knowles, 2017b; Shen et al., 2021). While general-purpose parallel translation data is abundantly available, automatically collecting a sufficient amount of domain-specific parallel data is challenging, and such translation for special purposes tends to require custom-made parallel data due to its specialized environment, e.g., terminologies in the medical domain, sometimes demanding a specialist to construct or check the quality of the parallel data. However, when we shift the focus from parallel data to monolingual data, it is possible to easily obtain such monolingual data for the target domain, and numerous pre-trained general NMT models have been developed.

In this study, we focus on leveraging pre-trained general NMT models that are easily accessible and attempt to transfer an NMT model pre-trained on a general domain into a domain-specific NMT model by using only the features obtained from the monolingual domain data of the translation target language. However, naively performing fine-tuning to alter the output of the pre-trained NMT model and forcibly changing the probability distribution can lead to catastrophic forgetting issues, ranging from the loss of fluency in translated sentences acquired during pre-training (Korbak et al., 2022; Choshen et al., 2020; Kiegeland & Kreutzer, 2021) to degradation in non-specific domains caused by overfitting to specific terminologies (Saunders & DeNeefe, 2024; Gu & Feng, 2020; Thompson et al., 2019), thereby causing a reduction in translation performance. To achieve the domain shift while reducing catastrophic forgetting by harmlessly modifying the model's knowledge to avoid degrading generalization performance or excessive overfitting to a specific domain, we represent the target domain as conditional energy-based models (EBMs) and approximate the EBMs using Conditional Distributional Policy Gradients (CDPG) (Korbak et al., 2022), which is a variant of the Generation under Distributional Control (GDC) framework (Khalifa et al., 2021).

Korbak et al. (2022) had only verified the effectiveness of CDPG for small shifts, such as translating numeral nouns (e.g., "two") as digits (e.g., "2"). We extend the framework by using the token-level statistics of the target domain as features and constructing a large number of EBMs, and approximating these to meet their constraints. Specifically, we shift the pre-trained NMT models toward the token-level unigram distribution of the target domain by CDPG, enabling domain shifts that better consider the frequency information of the entire target domain. As a result, we are able to scale CDPG to specific domains in a fine-grained manner and apply domain shift to the general NMT model without inducing catastrophic forgetting. We confirm its effectiveness in several domain adaptation benchmarks (Tian et al., 2014; Koehn & Knowles, 2017a; Aharoni & Goldberg, 2020) and scenarios, thus we achieved unsupervised domain adaptation using only target side domain data. Moreover, we proposed the DYNAMIC CDPG, which dynamically changes parameters using a small amount of bilingual validation data to select the best parameters, as a way to measure the upper-bound of our unsupervised domain adaptation. Analysis of the results of CDPG and DYNAMIC CDPG revealed that while selecting parameters sensitively can sometimes yield the best results, a simple CDPG can sufficiently achieve domain shift while reducing catastrophic forgetting.

## 2 CONDITIONAL DISTRIBUTIONAL POLICY GRADIENTS

Conditional Distributional Policy Gradients (CDPG) (Korbak et al., 2022) is a method that approximates the generative probabilities of a language model to a target distribution while preventing catastrophic forgetting. It softly modifies the pre-trained parameters $\theta$ by shifting the distribution slightly by EBMs through fine-tuning.

We define the pre-trained conditional language model $a(\boldsymbol{x}|\boldsymbol{c})$ where $\boldsymbol{c}$ is a context, i.e., an input source language sentence, and $\boldsymbol{x}$ is a sentence, i.e., in a target language, sampled from the entire distribution $\mathcal{X}$ given $\boldsymbol{c}$.

We introduce an energy-based model (EBM) $p_{\boldsymbol{c}}(\boldsymbol{x})$ as a controlled language model defined as:

$$p_{\boldsymbol{c}}(\boldsymbol{x}) = \frac{1}{Z_{\boldsymbol{c}}} a(\boldsymbol{x}|\boldsymbol{c}) b(\boldsymbol{x}, \boldsymbol{c}). \tag{1}$$

Here, $Z_{\boldsymbol{c}} = \sum_{\boldsymbol{x} \in \mathcal{X}} p(\boldsymbol{x}|\boldsymbol{c})$ is a partition function that normalizes the entire EBM $p_{\boldsymbol{c}}(\boldsymbol{x})$, and $b(\boldsymbol{x}, \boldsymbol{c})$ is a control condition function which is 1 when a certain constraint is met. When $b(\boldsymbol{x}, \boldsymbol{c})$ is reduced to a binary scorers $\phi_i(\boldsymbol{x}) \in \{0, 1\}$ as proposed by Khalifa et al. (2021), the EBM is formulated as:

$$p_{\boldsymbol{c}}^{point}(\boldsymbol{x}) = \frac{1}{Z_{\boldsymbol{c}}} a(\boldsymbol{x}|\boldsymbol{c}) \prod_i \phi_i(\boldsymbol{x}). \tag{2}$$

However, with binary constraints, only two values can be handled: either always meeting a specific condition or not, making it impossible to address needs such as satisfying a constraint with a probability of 0.5. For example, if we tackle to reduce the bias in the text generation style considering gender, the desired constraint is 0.5 female character and 0.5 male character. Khalifa et al. (2021) proposed a distributional constraint method for unconditional EBM $p(\boldsymbol{x}) = \frac{1}{Z} a(\boldsymbol{x}) b(\boldsymbol{x})$ to resolve the problem, and Kruszewski et al. (2023) adapt it to the conditional EBM with exponential family as follows:

$$p_{\boldsymbol{c}}^{dist}(\boldsymbol{x}|\boldsymbol{\lambda}) = \frac{1}{Z_{\boldsymbol{c}}} a(\boldsymbol{x}|\boldsymbol{c}) \exp(\boldsymbol{\lambda} \cdot \boldsymbol{\phi}(\boldsymbol{x}, \boldsymbol{c})), \tag{3}$$

where $\boldsymbol{\lambda}$ is a parameter vector of the distribution features. The parameter $\boldsymbol{\lambda}$ is determined through fine-tuning by starting from random initialization and iteratively updated by stochastic gradient descent (SGD) to minimize the loss function considering a distribution over contexts $\tau(\boldsymbol{c})$ as follows:

$$\nabla_{\boldsymbol{\lambda}} \mathcal{L}_{coef}(\boldsymbol{\lambda}) = \mathbb{E}_{\boldsymbol{c} \sim \tau(\boldsymbol{c})} \mathbb{E}_{\boldsymbol{x} \sim p_{\boldsymbol{c}}^{dist}(\cdot; \boldsymbol{\lambda})} \boldsymbol{\phi}(\boldsymbol{x}, \boldsymbol{c}) - \bar{\boldsymbol{\mu}}, \tag{4}$$

where $\bar{\boldsymbol{\mu}}$ is the probability for each feature and the moments $\mathbb{E}_{\boldsymbol{x} \sim p_{\boldsymbol{c}}^{dist}(\cdot; \boldsymbol{\lambda})}$ are computed through self-normalized importance sampling using $a(\cdot)$. In the previous example, if a female character is expected, the probability becomes 0.5.

However, since the EBM $p_{\boldsymbol{c}}(\boldsymbol{x})$ in Equation 1 that satisfies these constraints is not an autoregressive language model, it cannot perform generation. Therefore, training is conducted using the autoregressive model $\pi_{\theta}(\boldsymbol{x}|\boldsymbol{c})$ to approximate $p$ on average across contexts by minimizing the expected cross-entropy loss $\mathrm{CE}(\cdot)$ between $\pi_{\theta}(\boldsymbol{x}|\boldsymbol{c})$ and multiple $p_{\boldsymbol{c}}$ of the EBM as follows:

$$\mathcal{L}(\theta) = \mathbb{E}_{\boldsymbol{c} \sim \tau(\boldsymbol{c})} \mathrm{CE}\left(p_{\boldsymbol{c}}^{dist}(\cdot), \pi_{\theta}(\cdot \mid \boldsymbol{c})\right). \tag{5}$$

The gradient of this objective takes the following form:

$$\nabla_\theta \mathcal{L}(\theta) = \mathbb{E}_{\boldsymbol{c} \sim \tau(\boldsymbol{c})} \nabla_\theta \, \text{CE}\left(p_{\boldsymbol{c}}^{dist}(\cdot), \pi_\theta(\cdot \mid \boldsymbol{c})\right) \tag{6}$$

$$= -\mathbb{E}_{\boldsymbol{c} \sim \tau(\boldsymbol{c})} \mathbb{E}_{\boldsymbol{x} \sim p_{\boldsymbol{c}}^{dist}(\boldsymbol{x})} \nabla_\theta \log \pi_\theta(\boldsymbol{x} \mid \boldsymbol{c}) \tag{7}$$

$$= -\mathbb{E}_{\boldsymbol{c} \sim \tau(\boldsymbol{c})} \mathbb{E}_{\boldsymbol{x} \sim \pi_\theta(\boldsymbol{x} \mid \boldsymbol{c})} \frac{p_{\boldsymbol{c}}^{dist}(\boldsymbol{x})}{\pi_\theta(\boldsymbol{x} \mid \boldsymbol{c})} \nabla_\theta \log \pi_\theta(\boldsymbol{x} \mid \boldsymbol{c}). \tag{8}$$

The loss function is used by important sampling from $\pi_\theta$. By iteratively training these for $\theta$, $\pi_\theta$ can approximate the generative probability of the target EBM, enabling autoregressive generation. Details defer to Korbak et al. (2022). Note that the CDPG is a method for fine-tuning a model; thus, it does not introduce any changes to parameter size, model architecture, or inference speed.

## 3 DOMAIN ADAPTATION BY CDPG

### 3.1 ADAPTATION BY MONOLINGUAL FEATURES

Machine translation for a specific domain, e.g., medical domain, poses challenges for domain shifts and usually fine-tuing is required relying on high quality in-domain parallel data. However, creating such data might not be feasible especially when the rapid progress is happening in the domain, e.g., the development of new medicine reported by non-English documents. We leverage monolingual data in the specific domain in the target language, e.g., English reports in the medical domain, and propose domain adaptation for NMT with CDPG using only the subword frequency information as features so that domain specific terminologies and styles are reflected in NMT. When applying CDPG for NMT, the source sentence corresponds to a context $\boldsymbol{c}$, and the ideal target sentence is derived from $p_{\boldsymbol{c}}^{dist}(\boldsymbol{x}|\boldsymbol{\lambda})$. For training CDPG under distribution constraints, as shown in Equation 3, it requires a binary scorer $\phi_i(\boldsymbol{x}, \boldsymbol{c})$ and a parameter $\lambda_i$ for each feature.

To perform domain adaptation, we use as features whether each subword of the target domain is included in the output sentence, represented by $\phi(\boldsymbol{x}, \boldsymbol{c})$. Moreover, when learning the parameter vector $\boldsymbol{\lambda}$ according to Equation 4, we set the probability of each constraint, $\bar{\boldsymbol{\mu}}$, as the basis on the ratio of the frequency of subwords in the whole text in the target domain as follows:

$$\bar{\mu}_i = \frac{Freq^{target}(x_i)}{\sum_{x_j \in X} Freq^{target}(x_j)}, \tag{9}$$

where $Freq^{target}$ denotes the frequency of each subword $x_i$ in the target text in the vocabulary $X$. By performing the above operations, we attempt to address the domain shift by utilizing the frequency of all subwords of the target domain text. Since this feature selection only uses data from the target side, the creation of the EBM model only requires the target side domain text.

### 3.2 DYNAMIC CDPG

EBM is iteratively updated by Equation 4 to approximate the generative language model toward the expected probability distribution for the target domain. At this time, it generates multiple sentences $\boldsymbol{x}$ with context $\boldsymbol{c}$ through nucleus sampling (Holtzman et al., 2020). Specifically, the parameter of nucleus sampling, top-$p$, controls the diversity of generated outputs, where a lower value of top-$p$ means the generated sentences are closer to the target distribution. However, the initial distance between the distribution of the pre-trained model and the target distribution varies, meaning that CDPG requires different top-$p$ settings for different domains. Meanwhile, under the general settings of CDPG, the absence of a validation set prevents us from determining the top-$p$ value. Furthermore, the granularity at which the model approaches the target distribution in CDPG is not constant. Specifically, after a learning process with a given top-$p$ in CDPG, the model still preserves a distance from the target distribution, thus demanding a large top-$p$ value. Therefore, we introduce DYNAMIC CDPG that dynamically changes the top-$p$ in each iteration of the approximation to EBM in Equation 1 to investigate the upper-bound potential in applying CDPG with monolingual data.

A bilingual development set[1] is leveraged in DYNAMIC CDPG to guide the training process by measuring the current progress on the dataset. The basic idea of DYNAMIC CDPG is to divide the

---

[1]The development set refers to the text used to generate features.

training process into several iterations, then start with a constant parameter for top-$p$, and reconsider it in each training iteration such that a smaller top-$p$ will be selected in the next iteration if a larger top-$p$ leads to inferior performance on the development set. The detailed settings are described in Appendix B. Our preliminary studies showed that the training under DYNAMIC CDPG is always stable under our top-$p$ scheduling.

# 4 EXPERIMENTAL SETUP

## 4.1 DATASETS

We conduct experiments with four translation pairs of English to German (en→de), German to English (de→en), English to Chinese (en→zh), and Chinese to English (zh→en). For pairs involving de, we collect four domains, including IT, Medical, Law, and Koran from the public corpus[2] released by Koehn & Knowles (2017a); Aharoni & Goldberg (2020), where each domain has 2,000 sentences for the development set and test set, respectively. Given the low quality[3] of this corpus, we clean up and re-align the test set using de as the basis to avoid potential bias in evaluation. For pairs involving zh, we collect four domains, including Education, Laws, Thesis, and Science, from the UM-Corpus (Tian et al., 2014), which is public[4] with high quality. Although this corpus provides 456 – 790 sentences for test sets in those 4 domains, the development set is not provided. Therefore, we randomly select 3,000 sentences from the training data for each domain as the development sets. Moreover, we use the development sets[5] of WMT from 2018 – 2022, i.e., 14,482 translation instances of the newsdev set from a news domain, to train CDPG for all translation directions by treating them as a generic domain data set. Specifically, the contexts $\tau(c)$ are collected from the 14,482 source language sentences of the newsdev set and, the domain features $\bar{\mu}$ are derived from the target language sentences of the domain specific instances.

## 4.2 MODELS

We employ four open-source MT models (Tiedemann & Thottingal, 2020) from HuggingFace[6] as backbones in our experiments. Those models are based on Transformer (Vaswani et al., 2017) and are trained on OPUS with the same configuration[7] comprising the encoder and decoder layers of 6, attention heads of 8, embedding size of 512, inner size of 2048. Given that the fine-tuning of CDPG involves all parameters, we fine-tune models on the development sets as a baseline denoted by FINE-TUNED. Note that the back-translation (Sennrich et al., 2016) is not included as a baseline in our main experiments, because FINE-TUNED is based on real translation instances in the specific domains comprising a small number of sentences, e.g., only 3,000 instances each, representing the upper bound of the back-translation[8]. Furthermore, we employ LoRA for fine-tuning by adapting the attention weights (Hu et al., 2021) with the inner rank of 8 as the second baseline. All fine-tuning experiments are training for 10 epochs, and hyper-parameter settings are described in Appendix E. Finally, the checkpoint, which has the best performance on the development set, is measured for comparison. We used the *disco*[9] (Kruszewski et al., 2023) to implement the EBMs and the CDPG training code[10].

## 4.3 EVALUATION

We set the beam size of 4 for each model to generate translations for the entire test set, and did not employ nucleus sampling (Holtzman et al., 2020) in the final evaluation, because top-$p$ is the param-

---

[2]https://github.com/roeeaharoni/unsupervised-domain-clusters

[3]The low quality includes but is not limited to repetition, not alignment, and noise. Furthermore, the refined test data becomes unseen, enabling evaluation free from any data contamination issues in the existing training corpus (Raunak & Menezes, 2022). We will make the cleaned dataset publicly available for future studies.

[4]http://nlp2ct.cis.umac.mo/um-corpus/

[5]http://data.statmt.org/wmt23/general-task/dev.tgz

[6]https://huggingface.co/Helsinki-NLP

[7]Details in: https://hf.co/Helsinki-NLP/opus-mt-en-zh/blob/main/config.json

[8]We provide further details of the relationship between FINE-TUNED and back-translation in Appendix F.

[9]https://github.com/naver/disco

[10]The detailed implementation code for our experiments will be made available upon acceptance.

eter used only in the training process of CDPG. Then, translations are evaluated by four automatic MT evaluation methods: 1) Confidence (Müller et al., 2019; Wang et al., 2020), calculated by taking the average probability of each token at the generation[11], 2) BLEU (Papineni et al., 2002), assessed with the implementation of SacreBLEU (Post, 2018) to measure the surface-level similarities, 3) NIST (Doddington, 2002), which is similar to BLEU but gives special attention to low-frequency words to assess the qualities of domain-specific terminologies, and 4) BERTScore (Zhang et al., 2020), which reports embedding similarities by Precision, Recall, and F1 scores, where the F1 score being the harmonic mean of Precision and Recall[12]. Moreover, the statistical significance testing (Koehn, 2004) is conducted using paired bootstrap resampling with 1,000 iterations and 0.5 resampling ratios, where $p < 0.1$ means the difference is significant.

## 5 EXPERIMENTAL RESULTS

### 5.1 MAIN RESULTS

Table 1 shows the experimental results. First, FINE-TUNED and LORA fail to achieve improvement, except in *Medical* of en→de, *Laws* of en→zh, and *Thesis* and *Science* of zh→en, where they achieved slight enhancements. Second, even though CDPG are always improved in confidence, CDPG has a heavy fluctuation in its performances. Specifically, we observed gains in some domains, such as *IT* of en→de and *Education* of en→zh, comparable results with PRE-TRAINED on some domains, and degraded performance on others based on the assessments of the general evaluation methods. However, NIST scores, which give special attention to low-frequency words, of CDPG are still improved in those degraded domains. For instance, although CDPG demonstrates decreases of 0.92, 0.07, 0.05, and 0.05 in BLEU, P, R, and F1 scores, respectively, in the performance of *Laws* of zh→en, its NIST score achieves the improvement of 0.07, which is significantly better than PRE-TRAINED. The similar phenomena are also shown in *Medical* and *Koran* of en→de and *Medical* and *Law* of de→en. This result demonstrates that the high confidence in our methods arises from the improvement of the preference of models on domain-specific words, which are ignored by general automatic evaluation methods due to the relatively low frequency.

On the other hand, DYNAMIC CDPG shows the upper bound of the improvements of CDPG by guiding the training process on the bilingual development set. In the *Laws* of zh→en, it achieves the highest improvement, with specific gains in BLEU, P, R, and F1 scores of 3.01, 0.17, 0.12, and 0.31, respectively. Moreover, DYNAMIC CDPG also alleviates the extent of degradation to maintain the same level as with PRE-TRAINED, such as *Medical* of de→en and *Science* of zh→en. Notably, DYNAMIC CDPG is ineffective for the degradation in some cases, such as *Laws* of en→zh. Table 2 shows what top-$p$ is used in the training of DYNAMIC CDPG. Considering the results from Table 1, we observe that setting larger values for top-$p$ results in a minor increase in the confidence of models. For instance, setting them to 1 does not enhance confidence, and setting smaller values for top-$p$ leads to a more confident model. However, higher confidence does not lead to performance improvements. This observation leads to a hypothesis that the difference between the features used in CDPG and the original knowledge of the base model affects the final performance of CDPG.

### 5.2 WHEN IS CDPG EFFECTIVE?

Given the fluctuations in the performance of CDPG in Table 1, we will investigate the root cause of the problem. Specifically, we validate the hypothesis regarding the distributional differences presented in Section 5.1 by exploring the relationship between the features and the pre-trained models.

---

[11]The probability of generated tokens in an MT system is calculated by the Softmax function.

[12]Note that we did not include modern neural fine-tuned metrics, such as COMET (Rei et al., 2020b) and BLEURT (Sellam et al., 2020), as part of our main evaluation. These metrics are fine-tuned on human-generated MT quality annotation data (Ma et al., 2019), but such data does not capture sensitive patterns, such as named entity differences (Amrhein & Sennrich, 2022; Glushkova et al., 2023). Moreover, due to overfitting on the annotation data, these metrics tend to favor results closer to in-domain data of their fine-tuning data (Zouhar et al., 2024a;b). Consequently, we determined that such fine-tuned metrics are not suitable for domain adaptation experiments. Nonetheless, we included an evaluation with COMET in Appendix H. The results align with previous reports (Zouhar et al., 2024b; Amrhein & Sennrich, 2022) and we provide additional findings.

Table 1: Scores of our experiments. PRE-TRAINED indicates the performance of original models without fine-tuning. CDPG is trained by monolingual features only with 0.5 of top-$p$, and DYNAMIC CDPG is supervised by the bilingual development set. Conf. is the abbreviation of Confidence; P and R mean Precision and Recall scores of BERTScore, respectively. Lang. indicates the language involved in this pair, specifically, en→x and x→en indicate that translating from English and translating to English, respectively. The best score in each block, which is divided by the domain and pair, is in bold. Moreover, the decoration of † on the best score means it is significantly better than PRE-TRAINED and baselines according to the significance test with $p < 0.1$.

| Lang. | Domain | Method | en→x | | | | | | x→en | | | | | |
|---|---|---|---|---|---|---|---|---|---|---|---|---|---|---|
| | | | Conf. | BLEU | NIST | P | R | F1 | Conf. | BLEU | NIST | P | R | F1 |
| de | IT | PRE-TRAINED | 68.39 | 27.58 | 5.97 | 87.48 | 87.70 | 87.52 | 72.02 | 38.80 | 7.96 | 94.93 | 94.92 | 94.91 |
| | | FINE-TUNED | 67.91 | 27.92 | 6.04 | 87.38 | 87.60 | 87.42 | 71.76 | 38.83 | 7.95 | 94.94 | 94.93 | 94.92 |
| | | LoRA | 67.79 | 26.88 | 5.83 | 87.33 | 87.56 | 87.37 | 71.46 | 38.32 | 7.86 | 94.92 | 94.91 | 94.91 |
| | | CDPG | 74.44 | 29.01 | 6.25 | 87.68 | 87.77 | 87.67 | **77.91** | 39.79 | 8.30 | 94.95 | 94.94 | 94.93 |
| | | DYNAMIC CDPG | **79.36** | **30.78†** | **6.58†** | **88.00†** | **87.87** | **87.89†** | 77.65 | **40.55†** | **8.34†** | **95.01** | **94.96** | **94.98** |
| | Medical | PRE-TRAINED | 75.93 | 43.19 | 8.45 | 91.55 | 91.17 | 91.31 | 78.06 | **45.50** | 8.47 | **96.65** | **96.50** | **96.57** |
| | | FINE-TUNED | 75.71 | 43.23 | 8.46 | 91.53 | 91.14 | 91.29 | 77.77 | 45.48 | 8.47 | 96.64 | **96.50** | 96.56 |
| | | LoRA | 75.50 | **43.56** | 8.52 | 91.55 | 91.15 | 91.30 | 77.72 | 44.31 | 8.35 | 96.61 | 96.49 | 96.54 |
| | | CDPG | 80.85 | 42.54 | **8.60** | **91.61** | **91.28** | **91.40** | **82.84** | 44.56 | **8.56** | 96.57 | **96.50** | 96.53 |
| | | DYNAMIC CDPG | **82.32** | 43.51 | 8.54 | 91.60 | 91.20 | 91.36 | 77.72 | 45.06 | 8.55 | 96.63 | 96.47 | 96.54 |
| | Law | PRE-TRAINED | 72.49 | 44.82 | 9.01 | 89.38 | 89.11 | 89.22 | 72.89 | **51.75** | 10.05 | 96.06 | **95.75** | **95.90** |
| | | FINE-TUNED | 72.08 | 44.83 | 9.01 | 89.39 | 89.10 | 89.22 | 72.53 | 51.70 | 10.04 | 96.06 | 95.74 | 95.89 |
| | | LoRA | 72.05 | 44.80 | 9.01 | **89.42** | 89.12 | **89.25** | 72.55 | 51.67 | 10.04 | 96.05 | 95.73 | 95.89 |
| | | CDPG | 77.36 | 44.12 | **9.05** | 89.33 | **89.17** | 89.22 | **78.12** | 51.61 | 10.12 | 96.02 | 95.72 | 95.86 |
| | | DYNAMIC CDPG | **78.18** | **44.87** | 9.03 | 89.40 | 89.09 | 89.22 | 73.02 | 51.64 | **10.15** | **96.07** | 95.73 | 95.89 |
| | Koran | PRE-TRAINED | 61.51 | **18.90** | 5.25 | 81.59 | **80.18** | 80.84 | 59.23 | 20.86 | 5.66 | **91.95** | 91.07 | **91.49** |
| | | FINE-TUNED | 61.39 | 18.86 | 5.24 | 81.56 | 80.16 | 80.82 | 58.80 | 20.81 | 5.65 | 91.94 | 91.06 | 91.48 |
| | | LoRA | 61.18 | 18.86 | 5.24 | 81.54 | 80.13 | 80.80 | 58.94 | 20.83 | 5.65 | 91.94 | 91.05 | 91.48 |
| | | CDPG | **67.00** | 18.40 | **5.26** | 81.46 | 80.06 | 80.72 | **64.75** | **20.94** | **5.67** | 91.90 | **91.09** | 91.48 |
| | | DYNAMIC CDPG | 61.30 | 18.85 | 5.25 | **81.63** | 80.16 | **80.85** | **64.75** | **20.94** | **5.67** | 91.90 | **91.09** | 91.48 |
| zh | Education | PRE-TRAINED | 49.88 | 30.26 | 0.73 | 83.82 | 82.18 | 82.94 | 60.15 | 23.49 | 5.56 | 94.44 | 94.16 | 94.30 |
| | | FINE-TUNED | 49.28 | 30.07 | 0.68 | 83.70 | 81.96 | 82.78 | 59.63 | 23.54 | 5.56 | 94.43 | 94.16 | 94.29 |
| | | LoRA | 49.03 | 30.19 | 0.68 | 83.70 | 81.92 | 82.75 | 59.64 | 23.69 | 5.57 | 94.49 | 94.16 | 94.30 |
| | | CDPG | **57.88** | 31.03 | 0.93 | 84.59 | **83.23** | **83.86†** | 66.05 | 23.69 | 5.60 | 94.52 | **94.28** | **94.40** |
| | | DYNAMIC CDPG | 57.22 | **31.16†** | **0.94†** | **84.71†** | 83.01 | 83.81 | **67.02** | 24.23 | 5.67 | 94.60 | **94.28** | 94.28 |
| | Laws | PRE-TRAINED | 62.06 | 51.73 | 0.59 | 89.67 | **89.70** | 89.65 | 63.84 | 32.36 | 6.11 | 94.55 | 93.52 | 94.02 |
| | | FINE-TUNED | 61.46 | 51.71 | 0.59 | 89.74 | **89.70** | **89.69** | 63.47 | 32.27 | 6.10 | 94.52 | 93.49 | 93.99 |
| | | LoRA | 61.38 | **51.87** | 0.60 | **89.75** | 89.63 | 89.66 | 63.16 | 32.33 | 6.09 | 94.51 | 93.45 | 93.97 |
| | | CDPG | **68.50** | 50.81 | **0.68†** | 89.60 | 89.65 | 89.60 | 70.09 | **35.37†** | **6.45†** | **94.74†** | **93.95†** | **94.33†** |
| | | DYNAMIC CDPG | **68.50** | 50.81 | **0.68†** | 89.60 | 89.65 | 89.60 | 70.09 | **35.37†** | **6.45†** | **94.74†** | **93.95†** | **94.33†** |
| | Thesis | PRE-TRAINED | 47.62 | 18.95 | 1.14 | 76.09 | 75.69 | 75.78 | 50.83 | 8.65 | 3.48 | 89.55 | 88.33 | 88.92 |
| | | FINE-TUNED | 47.23 | 19.94 | 1.39 | 76.42 | **75.75** | 75.99 | 50.11 | 8.60 | 3.46 | 89.56 | 88.31 | 88.91 |
| | | LoRA | 47.22 | 19.34 | 1.25 | 76.36 | 75.72 | 75.93 | 50.15 | **8.71** | 3.48 | 89.58 | 88.33 | 88.93 |
| | | CDPG | **54.19** | 19.94 | 1.29 | 76.11 | 75.53 | 75.72 | 57.16 | 8.53 | 3.51 | 89.52 | 88.38 | 88.93 |
| | | DYNAMIC CDPG | 51.22 | **20.14** | **1.52†** | **76.53** | 75.72 | **76.03** | **58.57** | 8.49 | **3.54** | **89.67** | **88.37** | **89.00** |
| | Science | PRE-TRAINED | 47.56 | 24.45 | 0.94 | 81.28 | 79.06 | 80.09 | 57.97 | 16.20 | 4.86 | 92.80 | 92.60 | 92.69 |
| | | FINE-TUNED | 47.00 | 24.52 | 0.94 | 81.26 | 79.05 | 80.07 | 57.48 | **16.36** | **4.88** | **92.82** | 92.60 | **92.70** |
| | | LoRA | 46.75 | 24.57 | 0.96 | 81.38 | 79.09 | 80.15 | 57.49 | 16.29 | **4.88** | 92.81 | 92.60 | **92.70** |
| | | CDPG | **56.27** | 24.78 | **1.02** | 81.48 | **79.70†** | **80.53†** | 64.06 | 15.96 | 4.88 | 92.76 | **92.66** | **92.70** |
| | | DYNAMIC CDPG | 52.38 | **24.80** | 1.00 | **81.63†** | 79.39 | 80.43 | **65.55** | 16.34 | 4.85 | 92.79 | 92.60 | 92.69 |

Table 2: The top-$p$ values used in DYNAMIC CDPG. Those values are presented in the order they are used.

| | IT | Medical | Law | Koran |
|---|---|---|---|---|
| en→de | 0.5,0.4,0.8 | 0.5,0.7,1.0 | 0.5,0.8 | 1.0 |
| de→en | 0.5,0.9 | 1.0 | 0.5,0.9 | 0.5 |

| | Education | Laws | Thesis | Science |
|---|---|---|---|---|
| en→zh | 0.5,0.9 | 0.5 | 0.5,0.7 | 0.5,0.6,0.7 |
| zh→en | 0.5,0.4 | 0.5 | 0.5,0.6,0.7,0.8 | 0.5,0.3,0.2,0.1 |

First, following the process described in Section 3.1, we acquire features, i.e., expectations for binary scorers, from the development set denoted by *Dev Features*. Similarly, we obtain *Test Features*

Table 3: Comparisons on features. itr and uni are abbreviations of intersection and union, respectively; sim indicates similarity computed by the cosine similarity.

| Pair | Domain | Case of (i) | | Case of (ii) | |
| --- | --- | --- | --- | --- | --- |
| | | sim.itr (%) | sim.uni (%) | sim.itr (%) | sim.uni (%) |
| en→zh | Thesis | 74.88 | 73.23 | 92.81 | 91.82 |
| en→zh | Laws | 68.11 | 64.99 | 29.43 | 24.64 |
| zh→en | Education | 80.64 | 79.92 | 70.37 | 65.13 |
| zh→en | Science | 61.38 | 60.64 | 65.3 | 56.79 |
| en→de | IT | 83.14 | 65.09 | 93.14 | 90.99 |
| en→de | Koran | 95.69 | 95.48 | 98.81 | 98.67 |
| de→en | Law | 98.80 | 98.66 | 98.19 | 97.91 |
| de→en | Medical | 95.83 | 94.97 | 94.22 | 93.48 |

Table 4: This table shows the results of experiments on CDPG with different hyperparameters and corresponds to Table 3 row by row. Abbreviations in this table are consistent with Table 1. The best score in each row is in bold.

| Direction | Domain | top-$p$=0.5 | | | | top-$p$=0.8 | | | | top-$p$=1.0 | | | |
| --- | --- | --- | --- | --- | --- | --- | --- | --- | --- | --- | --- | --- | --- |
| | | Conf. | BLEU | NIST | F1 | Conf. | BLEU | NIST | F1 | Conf. | BLEU | NIST | F1 |
| en→zh | Thesis | **54.19** | 19.94 | 1.29 | 75.72 | 53.93 | **19.98** | **1.48** | **75.86** | 46.96 | 19.95 | 1.47 | 75.76 |
| en→zh | Laws | 68.50 | 50.81 | **0.69** | 89.60 | **68.78** | 51.16 | 0.65 | 89.63 | 61.68 | **51.90** | 0.61 | **89.71** |
| zh→en | Education | **66.05** | 23.69 | 5.59 | **94.40** | 65.86 | **23.92** | **5.65** | 94.37 | 59.68 | 23.50 | 5.58 | 94.31 |
| zh→en | Science | **64.06** | 15.96 | 4.81 | **92.70** | 63.93 | 16.14 | 4.87 | **92.70** | 57.29 | **16.34** | **4.88** | 92.69 |
| en→de | IT | 74.44 | 29.01 | 6.25 | **87.67** | **74.67** | **29.13** | **6.28** | 87.66 | 67.87 | 28.19 | 6.08 | 87.47 |
| en→de | Koran | 67.00 | 18.40 | 5.14 | 80.72 | **67.14** | 18.50 | 5.19 | 80.74 | 61.30 | **18.85** | **5.25** | **80.85** |
| de→en | Law | 78.12 | **51.61** | 10.12 | 95.86 | **78.33** | 51.53 | **10.16** | 95.86 | 71.83 | 51.58 | **10.16** | **95.87** |
| de→en | Medical | 82.84 | 44.56 | 8.43 | 96.53 | **83.06** | 44.82 | 8.47 | **96.54** | 77.72 | **45.06** | 8.47 | **96.54** |

from the test set. Subsequently, we generate translations on the development set using the pretrained model and derive features from translations denoted by *Pretrained Features*. We use the cosine similarity to compute the similarity between two sets of features: The case of (i) compares *Dev Features* and *Pretrained Features* to demonstrate that when does CDPG make models more confident; The case of (ii) compares *Dev Features* and *Test Features* to demonstrate that when is CDPG effective. Additionally, considering the different lengths of each feature set, we compare both the intersection and union of these sets.

Table 3 presents the analysis of features[13] to complement Tables 1 and 2. First, we observe that DYNAMIC CDPG encourages the model to align with *Dev Features* only when there is a low similarity between *Dev Features* and *Pretrained Features*. Specifically, in the process of DYNAMIC CDPG, the model would use lower top-$p$ values to increase the confidence of models. For instance, the similarity of the intersection and union for the *Thesis* of en→zh is 74.88 and 73.23, respectively, with top-$p$ values of 0.5 and 0.7, resulting in a confidence increase of 3.60. Conversely, when the similarity is high, DYNAMIC CDPG tends to preserve the knowledge of the pre-trained models. For example, the similarity for *Koran* of en→de is 95.69 and 95.48, with top-$p$ values of 1.0, leading to no increase in confidence. Furthermore, we find that the similarity between *Dev Features* and *Test Features* impacts the effectiveness of our approach. For instance, the similarity for *Laws* of en→zh is 29.43 and 24.64, indicating a significant difference between the features used in CDPG and the features of the test set. As a result, the performance degrades notably as reported in Table 1, even though the top-$p$ value is 0.5 and the confidence increases by 6.44. This analysis validates our hypotheses in Section 5.1 and further demonstrates that the fluctuations in the performance of CDPG are caused by the differences of the distribution in domains.

To further support this statement, we conduct experiments on CDPG with a fixed value for top-$p$. Table 4, which is row-aligned with Table 3, shows the results of domains with 3 different settings,

---

[13]The full statistical results, including the length of features, intersection, and union, are shown in Appendix D.

Table 5: Instances for generated test sets of PRE-TRAINED and CDPG, we select a short sentence and a long sentence for `de` and `zh`, respectively. In #Changes, the numerator indicates how many sentences are changed in the generated test texts of CDPG compared to PRE-TRAINED, and the denominator indicates the size of the test set. Underline means the translation is inaccurate. Words in red mean hitting the term accurately, but, words in blue mean that they are updated, but do not hit the target.

| Domain: Education | Pair: en→zh | #Changes: 408/790 |
|---|---|---|
| Input | What an absurd suggestion! | |
| Reference | 多荒谬的建议啊！ | |
| PRE-TRAINED | 胡说八道！ | |
| CDPG | 多么荒谬的建议！ | |

| Domain: Thesis | Pair: en→zh | #Changes: 414/625 |
|---|---|---|
| Input | Newton's transformation family f w(z)=z-1wz w-1 containing only one complex parameter w(w≠0 or 1) is constructed from the transcendental mapping z→e z w+c. 用超越复映射F(z) =ezw+c构造出含有单参数w(w≠ 0或1)的牛顿变换族fw(z) =z- 1wzw-1模型，fw(z)有可数无穷多个极值点。 | |
| Reference | | |
| PRE-TRAINED | 牛顿的变换型fw(z) =z-1wz W-1 仅包含一个复合参数w(w)0 或1) 的f(z) =z-1wz W-1。 | |
| CDPG | 牛顿的变换型fw(z) =z-1wz W-1 仅包含一个复合参数w(w)0 或1)，是用超常绘图ze z+c 构造的w-1模型。 | |

| Domain: IT | Pair: en→de | #Changes: 662/2000 |
|---|---|---|
| Input | SubDialog has one state, default. | |
| Reference | SubDialog hat nur einen Status, Standard. | |
| PRE-TRAINED | SubDialog hat einen Zustand, default. | |
| CDPG | SubDialog hat einen Zustand, Standard. | |

| Domain: Medical | Pair: en→de | #Changes: 748/2000 |
|---|---|---|
| Input | 4 ml of solution in a 5 ml vial (type I glass) closed with a latex-free stopper (bromobutyl/ isoprene polymer) and a seal (lacquered plastic). | |
| Reference | 4 ml Lösung in einer 5 ml-Durchstechflasche (Glastyp I), die mit einem latexfreien Stopfen (Bromobutyl/Isoprenpolymer) und eine Kappe (lacquered Kunststoff) verschlossen ist. | |
| PRE-TRAINED | 4 ml Lösung in einer 5-ml-Durchstechflasche (Glas Typ I), die mit einem latexfreien Stopfen (Brombutyl/Isoprenpolymer) und einem Siegel (Lackkunststoff) verschlossen ist. | |
| CDPG | 4 ml Lösung in einer 5 ml Durchstechflasche (Glas Typ I), die mit einem latexfreien Stopfen (Brombutyl/Isoprenpolymer) und einem Siegel (lackierter Kunststoff) verschlossen ist. | |

and the results follow the analysis of Table 3.[14] We categorize these results into two scenarios. First, when the similarity between *Dev Features* and *Pretrained Features* is low, once the similarity between *Dev Features* and *Test Features* is high, CDPG benefits with smaller parameters, as seen in the *Thesis* of en→zh and *IT* of en→de. Conversely, a parameter of 1 ensures the model's performance, such as Laws of en→zh and *Science* of zh→en. Subsequently, when the similarity between *Dev Features* and *Pretrained Features* is high, the enhancement from CDPG is always limited, thus showing minimal fluctuation and 1 is the safer parameter. Finally, we also observe that the confidence relates solely to the parameters. These results not only validate our hypothesis in Section 5.1, that the performance of CDPG is related to the provided monolingual features, but also demonstrate that even if CDPG effectively alters the knowledge of the base model, it may not be detected by the test set.

# 6 DISCUSSION

## 6.1 QUALITATIVE ANALYSIS

Given that the test set may not be able to accurately reflect the effect of CDPG, we conduct qualitative analysis to quantify the results in detail. Table 5 presents 4 translation instances. We first observe that CDPG only partially modifies the original model's knowledge demonstrated by only marginal changes in translations. Moreover, CDPG primarily enhances the model in word selection. Specifically, for two instances of en→de, regardless of sentence length, only keywords are changed without affecting the semantics and syntax, resulting in that not all inferences of the test set are changed. These findings confirm our motivation that CDPG can harmlessly modify the knowledge of models. Notably, these findings also explain the non-significant difference in BERTScore in Table 1, because representation-level evaluation methods are not sensitive to the word-specific changes.

---

[14] We illustrate experiments with parameters from 0.3 to 1.0, which are provided in Appendix C.

Table 6: Relative differences between scores of FINE-TUNED and scores of DYNAMIC CDPG. The second column and second row indicate the domain used for training and testing, respectively. Underline denotes that the value is in the aligned case, namely, training and testing are in the same domain. Gen.f.t and Gen.d.c. indicate the difference between PRE-TRAINED and FINE-TUNED and the difference between PRE-TRAINED and DYNAMIC CDPG on a generic domain (testing on the newstest2020), respectively, which are pivots to measure the relative difference.

| | | Confidence | | | | | BLEU Scores | | | | |
|---|---|---|---|---|---|---|---|---|---|---|---|
| | | Education | Thesis | Science | Gen.f.t | Gen.d.c. | Education | Thesis | Science | Gen.f.t | Gen.d.c. |
| →zh | Education | _7.94_ | 8.29 | 9.21 | -1.16 | 8.52 | _1.09_ | 0.29 | -0.44 | -0.76 | -0.17 |
| | Thesis | 6.70 | _3.99_ | 5.58 | -0.31 | 7.69 | 0.87 | _0.20_ | 0.37 | -0.28 | 0.13 |
| | Science | 4.83 | 4.20 | _5.38_ | -0.68 | 4.92 | 0.87 | 0.50 | _0.28_ | -0.02 | 0.33 |
| →en | Education | _7.39_ | 7.86 | 7.83 | -0.59 | 8.31 | _0.69_ | -0.07 | -0.27 | -0.11 | 0.19 |
| | Thesis | 7.72 | _8.46_ | 8.03 | -0.51 | 8.84 | 0.66 | _-0.11_ | 0.09 | -0.04 | 0.20 |
| | Science | 7.81 | 8.51 | _8.07_ | -0.55 | 8.89 | 0.64 | -0.26 | _-0.02_ | -0.07 | 0.26 |
| | | IT | Medical | Koran | Gen.f.t | Gen.d.c. | IT | Medical | Koran | Gen.f.t | Gen.d.c. |
| →de | IT | _10.61_ | 8.97 | 9.90 | -0.22 | 12.75 | _2.86_ | 0.38 | -1.13 | -0.15 | -1.65 |
| | Medical | 8.32 | _6.61_ | 7.11 | -0.27 | 9.47 | 2.44 | _0.28_ | -0.63 | -0.07 | -0.90 |
| | Koran | 0.65 | 0.47 | _-0.09_ | -0.21 | -0.86 | 1.05 | 0.93 | _-0.01_ | -0.18 | -0.08 |
| →en | IT | _5.89_ | 6.17 | 6.76 | -0.22 | 10.38 | _2.72_ | -0.78 | -0.04 | -0.11 | -0.81 |
| | Medical | -1.02 | _-0.05_ | -0.82 | -0.29 | -1.50 | -0.65 | _-0.42_ | -0.11 | -0.06 | -0.18 |
| | Koran | 6.07 | 5.92 | _5.95_ | -0.20 | 8.28 | 0.97 | -0.91 | _0.13_ | -0.14 | -0.40 |

However, these findings do not mean CDPG benefits only the ability of word selection. For the instance of *Thesis* of en→zh, the PRE-TRAINED shows issues of semantic loss and repetitive generation, while CDPG complements the missing semantics and addresses the repetition. This improvement may be due to the enhanced confidence provided by GDC. Similarly, in the short sentence from en→zh, the original model tends to translate the source sentences into Chinese idioms, which do not fully align semantically with the source sentences, i.e., ignoring the semantics of the word "suggestion." In contrast, CDPG perfectly translates the keywords, indicating that GDC increases the attention of models on keywords.

In addition, given that CDPG acts as a soft constraint, its use of keywords is not always accurate. For example, in the long sentence of en→zh, the blue words represent an error in translation. This occurs because CDPG translates "transcendental" and "mapping" separately, and both words are present in the given features. This observation further corroborates our analysis in Section 5.2.

## 6.2 WILL OTHER DOMAINS BE INFLUENCED?

The primary goal of CDPG is to encourage the distribution of the pre-trained model to approach the expectations of given features. However, there exists a risk in less generalization to other domains due to the fitting to a single domain by CDPG. As shown in Table 6, we conduct experiments to measure the performance changes of DYNAMIC CDPG in crossing domains from two perspectives: 1) The relative difference between FINE-TUNED and DYNAMIC CDPG in experimented domains; 2) The changes of FINE-TUNED and DYNAMIC CDPG in the generic domain. First, FINE-TUNED consistently shows a decrease in both confidence and performance in the generic domain, whereas DYNAMIC CDPG achieves a significant increase in confidence in most cases, albeit with some fluctuations in performance. This indicates that the improvements by our method are generalized. While DYNAMIC CDPG shows higher ability in generalization compared to FINE-TUNED in most cases, there are two type exceptions: 1) The changes in confidence influence the generalization, since CDPG induces a global increase in confidence rather than domain-specific. However, this indirect influence is generally limited. Although, the highest degradation of BLEU scores brought by increasing confidence is 1.13 on *Koran* of en→de, DYNAMIC CDPG correspondingly gains 2.86 BLEU scores in *IT*, which is significantly better than FINE-TUNED. 2) The performance of the aligned case is lower than that of cross-domain performances, such as *Thesis* of en→zh and *Medical* of en→de, suggesting that *dev features* have a negative impact. These results once again corroborate our analysis in Section 5.2, that the effectiveness of CDPG is closely linked to the provided features. We also evaluated the robustness of multi-domain adaptation, which can also be regarded as noisy domain adaptation, in Appendix G and conducted a qualitative analysis of unseen

terminology domain adaptation in Appendix I. These results align with the strengths of our CDPG method.

## 7 RELATED WORK

When using parallel data, Luong & Manning (2015); Freitag & Al-Onaizan (2016) perform domain adaptation by training on large-scale general domain data, then fine-tuning on a small amount of domain data. Chu et al. (2017) mix general domain data and a small amount of domain data for training at once. Furthermore, efficient domain adaptation is aimed through the use of add domain tags (Kobus et al., 2017; Britz et al., 2017), considering subword tokenization units (Enomoto et al., 2023), and data sampling for training steps (Wang et al., 2017). However, direct fine-tuning with a small amount of data can lead to overfitting, so techniques like knowledge distillation (Dakwale & Monz, 2017) and regularization (Miceli Barone et al., 2017) are proposed.

When focusing on the utilization of monolingual data, some methods have been explored such as back translation (Sennrich et al., 2016), direct learning from monolingual data as LM (Gulcehre et al., 2015; Zhang & Zong, 2016; Domhan & Hieber, 2017; Burlot & Yvon, 2018), exploiting task-specific features (Dou et al., 2019b;a), utilizing knowledge graphs (Moussallem et al., 2019; Zhao et al., 2020), and nearest neighbor search (Farajian et al., 2017; Bapna & Firat, 2019; Zheng et al., 2021; Khandelwal et al., 2021; Wang et al., 2022; Deguchi et al., 2023; Agrawal et al., 2023), and the combination of unsupervised NMT methods and back-translation technique (Mahdieh et al., 2020). However, it can be challenging to find similar sentences in domain adaptation settings. Moreover, they rely on a large amount of monolingual data, but obtaining sufficient domain data is difficult.

For terminology constrained decoding, hard constrained decoding methods (Hokamp & Liu, 2017; Post & Vilar, 2018; Hu et al., 2019) by forcing the decoding of specific terminology, and soft constrained decoding methods Song et al. (2019); Chen et al. (2020) that use post-editing techniques using phrase tables are proposed. However, since these approaches require predefined constrained vocabularies, they face challenges when applied to real NMT scenarios that require inductive domain adaptation, such as handling unseen terminology.

The original paper of CDPG method (Korbak et al., 2022) which is used in our study, explores only minor changes such as converting numerical numbers to alphabetical numbers, not large-scale domain adaptation that considers the distribution of the entire target domain. About reinforcement learning methods (Ranzato et al., 2016; Kreutzer et al., 2017; Choshen et al., 2020; Kiegeland & Kreutzer, 2021; Yang et al., 2024), outside of the GDC framework, rewards are based only on overall scores such as BLEU, without the ability to impose fine-grained constraints. Furthermore, there is a potential for causing catastrophic forgetting, making scaling like in this study particularly challenging.

## 8 CONCLUSION AND FUTURE WORKS

We performed unsupervised domain adaptation by imposing large-scale distribution constraints using only features obtained from the entire target domain data through the CDPG method. Additionally, to effective large-scale constraints on CDPG, we proposed DYNAMIC CDPG, which dynamically changes feature selection in the training step, and verified its effectiveness. Although this experiment utilized a large-scale pre-trained NMT model, next, we aim to explore the potential of large-scale distribution constraints for cross-linguistic domain adaptation, such as improving translation performance for specific languages in low-resource languages or multilingual NMT models. In addition, in this study, we used the word distribution of the target domain as feature representations. However, we believe that exploring optimal feature selection, such as $n$-gram features or language model embeddings, for fine-tuning with CDPG should be pursued as a future direction.

### ETHICS STATEMENT

All datasets and models used in this work are public data, and we can use the data for research purposes. Moreover, there is no harmful content included in the examples used in the paper. Therefore, there are no ethical problems.

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

## A  LIMITATIONS

There are two main limitations in this work. The first is the limitation of our methodology, that is, although CDPG can accurately modify the knowledge of base models, the soft constraint of CDPG mentioned in Section 6.1 serves as both an advantage and a limitation. Specifically, several features used during training may correspond to the same semantics, in which case the final translation may not necessarily be the most ideal word from the perspective of human evaluation. The second is the limitation of the evaluation in our experiments. As the statements in Sections 5.1 and 6.1, representation-level evaluation MT methods are not sensitive to the improvements of CDPG, which not only results in the non-significant difference on BERTScore (Zhang et al., 2020). Moreover, even though NIST (Doddington, 2002) provides a reasonable assessment, NIST is limited by its BLEU style. Thus, exploring the awareness of representation-level evaluation methods on word-specific changes is considered as a future work.

## B  DETAILED SETTINGS OF DYNAMIC CDPG

For each iteration, we use an evaluation method, e.g., BLEU (Papineni et al., 2002), to assess the model's performance to decide whether to accept that iteration. Specifically, we heuristically define two potential value sets for top-$p$, $\mathbb{A} = [0.5, 0.4, 0.3, 0.2, 0.1]$ in descending order and $\mathbb{B} = [0.6, 0.7, 0.8, 0.9, 1.0]$ in ascending order, where $\mathbb{A}$ enables the model to gradually fit with the target features, while $\mathbb{B}$ implies gradually conservative behavior in learning by sampling diverse tokens. We start the iteration with the first element of $\mathbb{A}$ as the value of top-$p$; if this iteration is accepted, we proceed to the next iteration with the second element of $\mathbb{A}$; if rejected, we switch to $\mathbb{B}$ and continue iterating until all elements in either $\mathbb{A}$ or $\mathbb{B}$ are completely iterated.

## C  MORE GRANULAR EXPERIMENTS FOR VERIFYING HINTS

Figure 1 visualizes our experimental results including scores on the development set and scores on the test set. First, Figures 1a, 1b, 1c, and 1d show the confidence results for all 4 translation pairs. We find that changes in model confidence relate solely to the parameters. Subsequently, Figures 1e, 1f, 1g, and 1h sequentially present the results for *Koran* of en→de in terms of BLEU and BERTScore metrics. We observe that with high similarity between features (as indicated in Table 3), GDC performance decreases as parameter settings reduce. Finally, Figures 1i, 1j, 1k, and 1l show the results for *Thesis* of en→de. We note that when there is low similarity between *dev features* and *pretrained features*, performance on the development set improves with decreased parameter settings, although the trend on the test set does not completely follow the development set trend. These findings validate our statement in Section 5.2.

## D  FULL STATISTICAL RESULTS OF FEATURES

Full statistical results of features are shown in Table 7. We additionally provide the length of features extracted from each set, the length of the intersection, and the length of the union to show the comparison comprehensively.

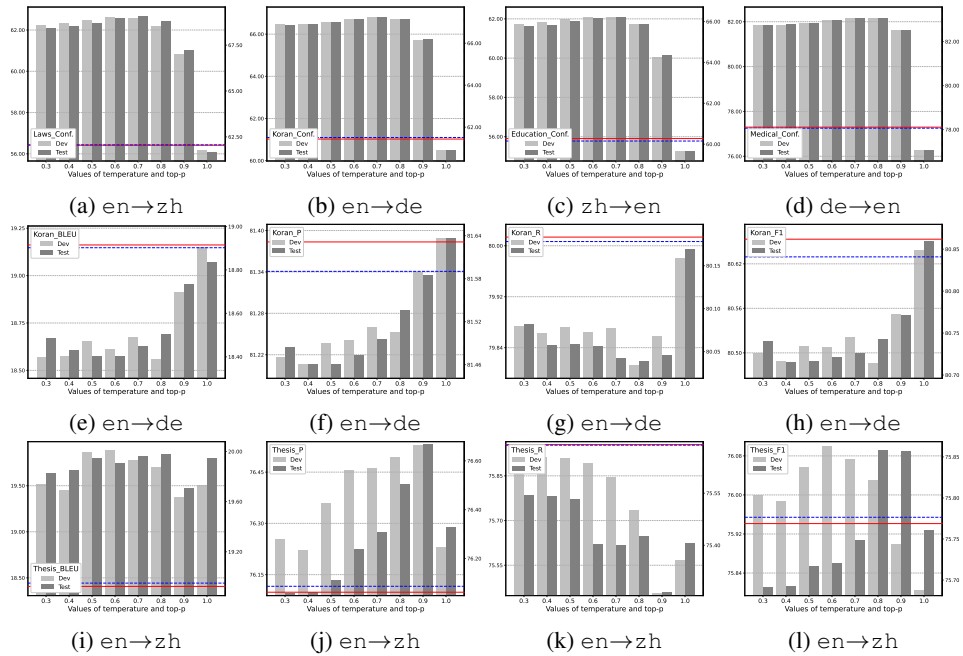

Figure 1: Illustrations of experimental results. For each subfigure, the caption shows the translation pair and the legend shows the domain and the metric. The left vertical axis is the score on the development set, the right axis is the score on the test set, and the horizontal axis is the top-$p$ values. In addition, the red and blue dashed lines are the scores of the PRE-TRAINED on the development set and the test set, respectively.

Table 7: Corresponding to Table 3. #len.1 means the length of features in the first set; itr and uni are abbreviations of intersection and union, respectively.

| Pair | Domain | Dev Features v.s. Pretrained Features | | | | | | Dev Features v.s. Test Features | | | | | |
|------|--------|--------|--------|---------|------------|---------|-------------|--------|--------|---------|------------|---------|-------------|
| | | #len.1 | #len.2 | #len.itr | sim.itr(%) | #len.uni | sim.uni(%) | #len.1 | #len.2 | #len.itr | sim.itr(%) | #len.uni | sim.uni(%) |
| en→zh | Thesis | 7533 | 7518 | 5395 | 74.88 | 9656 | 73.23 | 7533 | 3755 | 3188 | 92.81 | 8100 | 91.82 |
| en→zh | Laws | 6903 | 6783 | 4865 | 68.11 | 8821 | 64.99 | 6903 | 1852 | 1373 | 29.43 | 7382 | 24.64 |
| zh→en | Education | 10680 | 9546 | 7379 | 80.64 | 12847 | 79.92 | 10239 | 2357 | 1885 | 70.37 | 10711 | 65.13 |
| zh→en | Science | 9807 | 9127 | 6866 | 61.38 | 12068 | 60.64 | 10920 | 3089 | 2317 | 65.39 | 11692 | 56.79 |
| en→de | IT | 5832 | 5553 | 4152 | 83.14 | 7233 | 65.09 | 5832 | 5475 | 3366 | 93.14 | 7941 | 90.99 |
| en→de | Koran | 4543 | 3948 | 2931 | 95.69 | 5560 | 95.48 | 4543 | 4435 | 3300 | 98.81 | 5678 | 98.67 |
| de→en | Law | 7054 | 6469 | 5668 | 98.80 | 7855 | 98.66 | 7054 | 7014 | 4754 | 98.19 | 9314 | 97.91 |
| de→en | Medical | 6543 | 6130 | 5367 | 95.83 | 7306 | 94.97 | 6543 | 6577 | 4604 | 94.22 | 8516 | 93.48 |

# E    TRAINING DETAILS

**CDPG**    For training the parameter vector $\lambda$ in Equation 4, we set a batch size of 8 and a learning rate of 0.05 with a constant learning rate scheduler based on the training loss in our preliminary studies. Likewise, for fine-tuning CDPG model parameters $\theta$ in Equation 5, we set batch size of 128, epochs of 10, and learning rate of 2e-5 with a constant learning rate scheduler and Adam optimizer (Kingma & Ba, 2017). We always set top-$p$ to 0.5 in training $\lambda$ and fine-tuning $\theta$. Moreover, we set the character length of the considered features, i.e., subwords, to be no less than 3 to filter insignificant features, and the input texts are pre-processed by the tokenizer in each pre-trained model.

**Dynamic CDPG**    We maintain the hyperparameters of CDPG for DYNAMIC CDPG. We set each iteration of DYNAMIC CDPG to 10 epochs. We use both BLEU (Papineni et al., 2002) and BERTScore (Zhang et al., 2020) to calculate the validation score for each epoch. Additionally, we set a bar that requires at least three improvements in the validation score for an iteration to be accepted. Furthermore, the initial learning rate of subsequent iteration is set to dividing the initial

learning rate of the previously accepted iteration by the square root of the number of epochs to ensure training stability.

**Fine-tuning and LoRA**    We generally follow the original settings from the released checkpoints for FINE-TUNED, but we adjust the batch size to 128 and set the learning rate to 2e-7. We set the learning rate to 2e-7 for LORA.

## F    VERIFICATION OF BACK-TRANSLATION

In Section 4.2, we state that fine-tuning the model on bilingual data represents the upper bound of enhancement achievable through back-translation (Sennrich et al., 2016). Therefore, the back-translation results are not included in the main results, i.e., Table 1. In this appendix, we list the results of back-translation. Specifically, first, we generate source-language data using the corresponding reverse-direction model based on the data of the target language used in fine-tuning. We then fine-tune the model using the same settings on the generated data. The results are shown in Table 8.

## G    DESCRIPTION OF ROBUSTNESS

As shown in Table 9, we demonstrate the robustness of our method by comparing the performance trends of FINE-TUNED and CDPG in mixed-domain scenarios, in which an extra domain dataset is contaminated during training. The results reveal that the performance of FINE-TUNED consistently declines as the degree of domain mixing increases. In contrast, the performance of CDPG remains unaffected by the mixture of domains, underscoring its robustness.

## H    USAGE OF COMET

In our main experiments, we use BERTScore (Zhang et al., 2020) to measure the semantic similarity of inference results at the representation level. However, we do not include another popular representation-level metric, COMET (Rei et al., 2020a), in our main experiments due to observed irregularities in its results under certain cases. Specifically, as shown in Table 10, we notice that for translations involving German, COMET scores exhibit trends opposite to BLEU scores, with minimal score fluctuations. To investigate this phenomenon further, we conduct sentence-level analyses with the assistance of GPT-4o (OpenAI, 2024), as presented in Table 11. Overall, improvements in certain terms are evaluated negatively by *Unbabel/wmt22-comet-da*. A possible explanation for this behavior is that COMET emphasizes sentence-level coherence, which might conflict with domain-specific term adaptations in translations. In contrast, BERTScore, although also a representation-level metric, measures semantic similarity at the token level, making it more sensitive to term-level changes. It is worth noting that a deeper analysis of COMET's behavior lies beyond the scope of this work. Consequently, we choose to use BERTScore rather than COMET in this study.

## I    GENERALIZATION OF DOMAIN FEATURES

Table 12 shows two instances of en→de. As discussed in Section 5.1, CDPG tends to increase the confidence of the model. As a result, the inference of CDPG in Case #1 removes the repetition in PRE-TRAINED. Moreover, CDPG in Case #2 hits the feature in reference by fixing the original inaccurate word "Tunnelgeräts" to "Tunnelgerätes", which is not a feature used in fine-tuning. Namely, Case #2 shows the generalization of domain features in our proposed method. We therefore suspect that the essence of increasing confidence is to encourage the model to be close to the target domain.

Table 8: Scores of back-translation. BACK-TRANS indicates the model fine-tuned by the back-translation. Src and Tgt abbreviate the source language and the target language, respectively. All details follow Table 1.

| Src | Tgt | Domain | Method | Conf. | BLEU | P | R | F1 |
|-----|-----|--------|--------|-------|------|---|---|-----|
| en | zh | Education | PRE-TRAINED | 49.88 | **30.26** | **83.82** | **82.18** | **82.94** |
| | | | FINE-TUNED | 49.28 | 30.07 | 83.70 | 81.96 | 82.78 |
| | | | BACK-TRANS | 49.26 | 30.00 | 83.67 | 81.92 | 82.74 |
| | | Thesis | PRE-TRAINED | 47.62 | 18.95 | 76.09 | 75.69 | 75.78 |
| | | | FINE-TUNED | 47.23 | **19.94** | **76.42** | **75.75** | **75.99** |
| | | | BACK-TRANS | 47.20 | 19.30 | 76.41 | 75.70 | 75.95 |
| zh | en | Laws | PRE-TRAINED | 63.84 | **32.36** | **94.55** | **93.52** | **94.02** |
| | | | FINE-TUNED | 63.47 | 32.27 | 94.52 | 93.49 | 93.99 |
| | | | BACK-TRANS | 63.18 | 32.22 | 94.52 | 93.46 | 93.97 |
| | | Science | PRE-TRAINED | 57.97 | 16.20 | 92.80 | 92.60 | 92.69 |
| | | | FINE-TUNED | 57.48 | **16.36** | **92.82** | **92.60** | **92.70** |
| | | | BACK-TRANS | 57.48 | 16.33 | **92.82** | 92.59 | 92.69 |
| en | de | IT | PRE-TRAINED | 68.39 | 27.58 | **87.48** | **87.70** | **87.52** |
| | | | FINE-TUNED | 67.91 | **27.92** | 87.38 | 87.60 | 87.42 |
| | | | BACK-TRANS | 67.90 | 27.89 | 87.37 | 87.59 | 87.41 |
| | | Medical | PRE-TRAINED | 75.93 | 43.19 | **91.55** | **91.17** | **91.31** |
| | | | FINE-TUNED | 75.71 | **43.23** | 91.53 | 91.14 | 91.29 |
| | | | BACK-TRANS | 75.72 | 43.21 | 91.53 | 91.14 | 91.29 |
| de | en | Koran | PRE-TRAINED | 59.23 | **20.86** | **91.95** | **91.07** | **91.49** |
| | | | FINE-TUNED | 58.80 | 20.81 | 91.94 | 91.06 | 91.48 |
| | | | BACK-TRANS | 58.78 | 20.79 | 91.92 | 91.05 | 91.47 |
| | | Law | PRE-TRAINED | 72.89 | **51.75** | **96.06** | **95.75** | **95.90** |
| | | | FINE-TUNED | 72.53 | 51.70 | **96.06** | 95.74 | 95.89 |
| | | | BACK-TRANS | 72.53 | 51.71 | **96.06** | 95.74 | 95.89 |

Table 9: Scores of experiments on mixing data of two domains. The data in Domain is fixed, and we add sentences extracted from Mix.Domain into Domain. Then, we test the model performance in Domain. #Sent. indicates the number of added sentences. The best value in each block is in bold.

| Src | Tgt | Domain | Mix.Domain | Method | #Sent. | Conf. | BLEU | P | R | F1 |
|-----|-----|--------|-----------|--------|--------|-------|------|---|---|-----|
| en | de | IT | Medical | FINE-TUNED | 0 | 67.91 | **27.92** | **87.38** | **87.60** | **87.42** |
| | | | | | 500 | 67.82 | 27.63 | 87.35 | 87.57 | 87.39 |
| | | | | | 1000 | 67.75 | 27.61 | 87.36 | 87.58 | 87.40 |
| | | | | | 2000 | 67.61 | 27.27 | 87.32 | 87.55 | 87.36 |
| | | | | CDPG | 0 | 74.29 | 29.32 | **87.70** | 87.79 | 87.69 |
| | | | | | 500 | 74.24 | **29.70** | **87.70** | **87.80** | **87.70** |
| | | | | | 1000 | 74.22 | 28.83 | 87.63 | 87.77 | 87.64 |
| | | | | | 2000 | 74.14 | 29.43 | 87.68 | 87.79 | 87.68 |
| en | zh | Thesis | Laws | FINE-TUNED | 0 | 47.23 | **19.94** | 76.42 | **75.75** | **75.99** |
| | | | | | 750 | 47.14 | 19.77 | **76.44** | 75.73 | 75.98 |
| | | | | | 1500 | 47.05 | 19.63 | 76.42 | **75.75** | 75.98 |
| | | | | | 3000 | 46.83 | 19.13 | 76.37 | 75.74 | 75.95 |
| | | | | CDPG | 0 | 54.19 | 19.94 | 76.11 | 75.53 | 75.72 |
| | | | | | 750 | 54.16 | 20.06 | **76.25** | **75.59** | **75.81** |
| | | | | | 1500 | 54.12 | **20.15** | 76.21 | **75.59** | 75.80 |
| | | | | | 3000 | 54.01 | 20.10 | 76.24 | 75.58 | 75.80 |

Table 10: Scores of COMET, measured by *Unbabel/wmt22-comet-da*.

| Direction | Domain | Method | BLEU | COMET | Direction | BLEU | COMET |
|---|---|---|---|---|---|---|---|
| en→de | IT | PRE-TRAINED | 27.58 | 83.31 | de→en | 38.80 | 87.45 |
| | | FINE-TUNED | 27.92 | 83.24 | | 38.83 | 87.44 |
| | | CDPG | 29.32 | 83.38 | | 39.79 | 87.52 |
| | | DYNAMIC CDPG | 30.78 | 83.59 | | 40.55 | 87.56 |
| | Koran | PRE-TRAINED | 18.90 | 72.85 | | 20.86 | 73.92 |
| | | FINE-TUNED | 18.86 | 72.83 | | 20.81 | 73.90 |
| | | CDPG | 18.85 | 72.85 | | 20.94 | 73.84 |
| | | DYNAMIC CDPG | 18.85 | 72.85 | | 20.94 | 73.84 |
| | Law | PRE-TRAINED | 44.82 | 87.05 | | 51.75 | 87.11 |
| | | FINE-TUNED | 44.83 | 87.04 | | 51.70 | 87.09 |
| | | CDPG | 44.12 | 86.95 | | 51.64 | 87.13 |
| | | DYNAMIC CDPG | 44.87 | 86.92 | | 51.64 | 87.10 |
| | Medical | PRE-TRAINED | 43.19 | 87.79 | | 45.50 | 89.88 |
| | | FINE-TUNED | 43.23 | 87.76 | | 45.48 | 89.81 |
| | | CDPG | 42.54 | 87.74 | | 44.56 | 89.81 |
| | | DYNAMIC CDPG | 43.51 | 87.66 | | 45.06 | 89.81 |
| de→en | Education | PRE-TRAINED | 30.26 | 84.41 | zh→en | 23.49 | 82.99 |
| | | FINE-TUNED | 30.07 | 84.39 | | 23.54 | 83.02 |
| | | CDPG | 31.27 | 84.66 | | 23.59 | 83.38 |
| | | DYNAMIC CDPG | 31.16 | 84.65 | | 24.23 | 83.38 |
| | Laws | PRE-TRAINED | 51.73 | 89.45 | | 32.36 | 81.66 |
| | | FINE-TUNED | 51.71 | 89.43 | | 32.27 | 81.50 |
| | | CDPG | 51.90 | 89.69 | | 35.57 | 82.57 |
| | | DYNAMIC CDPG | 50.81 | 89.74 | | 35.57 | 82.57 |
| | Thesis | PRE-TRAINED | 18.95 | 70.62 | | 8.65 | 69.21 |
| | | FINE-TUNED | 19.94 | 70.58 | | 8.60 | 69.18 |
| | | CDPG | 19.94 | 70.89 | | 8.53 | 69.40 |
| | | DYNAMIC CDPG | 20.14 | 70.86 | | 8.49 | 69.47 |
| | Science | PRE-TRAINED | 24.45 | 78.80 | | 16.20 | 81.03 |
| | | FINE-TUNED | 24.52 | 78.78 | | 16.36 | 81.03 |
| | | CDPG | 24.94 | 79.38 | | 15.96 | 80.94 |
| | | DYNAMIC CDPG | 24.80 | 79.32 | | 16.34 | 80.88 |

Table 11: Bad cases of COMET. GPT-4o makes the Judgment.

| Input | Reference | Generation | Scores | Judgment |
|---|---|---|---|---|
| Screen only check box | Nur Bildschirm-Markierfeld | **PRE-TRAINED:** Nur Kontrollkästchen für den Bildschirm | 72.22 | CDPG |
| | | **CDPG**: Nur das Kontrollkästchen für den Bildschirm | 61.60 | |
| Failed to finalize | Fehler beim Finalisieren | **PRE-TRAINED:** Nicht fertig gestellt | 66.67 | CDPG |
| | | **CDPG**: Nicht abgeschlossen | 61.60 | |
| Enforce private variables to be private across modules by setting CompatibilityMode(true). | Durch das Setzen von CompatibilityMode(true) werden private Variablen bezüglich eines einzelnen Moduls als privat behandelt. | **PRE-TRAINED:** Private Variablen müssen über Module hinweg privat sein, indem Sie CompatibilityMode(true) einstellen. | 81.14 | CDPG |
| | | **CDPG**: Erzwingen Sie private Variablen, um über Module hinweg privat zu sein, indem Sie KompatibilitätMode (true) einstellen. | 68.47 | |

Table 12: Instances showing generalized features. Case #1 shows the direct influence of increased confidence. Case #2 shows the hitting of the target-domain feature, which is not included in fine-tuning features.

| Case #1 | |
| --- | --- |
| Input | PPM. |
| Reference | PPM. |
| PRE-TRAINED | PPM. - Nein, nein, nein, nein, nein, nein, nein, nein, nein, nein... |
| CDPG | PPM. |
| **Case #2** | |
| Input | This is the type of your tunnel device. |
| Reference | Dies ist der Typ des Tunnelgerätes. |
| PRE-TRAINED | Dies ist der Typ Ihres Tunnelgeräts. |
| CDPG | Dies ist der Typ Ihres Tunnelgerätes. |