# OpenReview forum: "Toward Domain Translation with Monolingual Domain Data Only"
_ICLR.cc/2025/Conference — Submitted to ICLR 2025_

### Official Review · Reviewer_n3SL · 2024-10-22

**Soundness:** 2
**Presentation:** 3
**Contribution:** 1
**Rating:** 3
**Confidence:** 4

**Summary:**

This paper addresses the topic of adapting a neural machine translation model to a specific domain when only target-language domain-specific data is available. The proposed approach uses energy-based models based on the vocabulary of the domain-specific data to adapt an existing NMT model to the specific data. The proposed model improves translation quality somewhat inconsistently, and through analysis this is attributed to improvements on domain-specific terminology.

**Strengths:**

1. The proposed method is described clearly and the method itself is well-justified.

2. There is a fair amount of analysis into successful and failure cases.

3. The results include statistical significance testing.

**Weaknesses:**

1. This paper would benefit from clearly defining and justifying the task at hand. It addresses domain adaptation with target-side monolingual data, and that is well justified. However, there are several implicit assumptions throughout that do not seem to be stated clearly or given a justification. For example, a) assuming that they are starting from a generic trained model to adapt, and cannot train a model to the relevant domain from scratch (section 4.2), b) assuming that the adapted model shouldn't have catastrophic forgetting of the original domain (line 45), c) assuming that it is desirable to do well on tasks that were unseen in both the original model and the adaptation data (table 6), and d) that the adapted output should exhibit minimal changes compared to the original model (line 410). There is nothing wrong with these assumptions necessarily, but you need to clearly state them and justify why you are restricting your exploration to these.

2. There are some issues with the evaluations. First, the base model that is used is OPUS-MT, but the domain-specific datasets that are used for evaluation for EN<->DE come from OPUS, so they were used to train the base model. Thus, this is not a true scenario of domain adaptation to an unseen domain, but one of domain shift. It is not clear to me whether this was done intentionally, but I think it would be preferable to do some domain adaptation evaluations with unseen data (and this might explain the lack of consistent positive results for any of the domain adaptation models, including the baselines and the EBM approach). Second, confidence is used as an evaluation score, when it is not clear that this correlates with any sort of meaningful MT evaluation. Third, the examples given in table 5 point more towards overfitting to the vocabulary of a specific dataset (*not* a specific domain) than to any true translation quality improvements.

3. This paper should take a broader view of the literature, including terminology-constrained machine translation (which seems to be hinted at as the ultimate goal of the proposed approach, e.g. in line 410 and table 5) as well as cases where the assumptions I listed in item 1 are relaxed (e.g., training a domain-specific model from scratch). In addition, the following paper is directly related (even without taking a broader view) and should be used as a baseline: https://arxiv.org/pdf/2010.12652.

4. Beyond the missing citations, the baselines are insufficient or problematic. a) The proposed approach should be compared against LLM translation, both generic and using in-context learning with monolingual examples, the latter of which would directly address the problem at hand. b) The fine-tuning comparisons only evaluate i) fine-tuning the entire model and ii) fine-tuning only the attention weights. To me given the small dataset and focus on target-side data it would make sense to explore other approaches like fine-tuning the decoder only. c) Line 199 says "the checkpoint, which has the best performance on the development set, is measured for comparison." but line 193 says fine-tuning is done on the development set. So the models are fine-tuned on the same set that is used for checkpoint selection; it would not be surprising if they don't generalize well to the test set.

**Questions:**

1. In line 36, you say "automatically collecting a sufficient amount of domain-specific parallel data is challenging". It would be good to get some quantitative information to justify this statement, particularly what you mean by "sufficient". In general, fine-tuning and ICL can work well with an extremely small corpus.

2. Line 44 says: "However, naively performing fine-tuning [...] can lead to catastrophic forgetting issues, such as the loss of fluency in the translated sentences acquired during pre-training, thereby causing a reduction in translation performance". Can you share evidence of this? Typically, catastrophic forgetting doesn't cause a loss of *fluency* in NMT per se, but just poorer performance on seen domains.

3. It would be good to add a discussion of whether the EBMs increase the parameter size, memory footprint, or inference speed of the model.

4. Line 194 should cite the back-translation paper. Also, I would recommend including a comparison to back-translation as a baseline, and labeling "fine-tuned" as your upper bound.

5. I found the presentation of table 6 extremely confusing. It would be clearer to simply show the BLEU scores of the two models, rather than showing the difference between them. In addition, if you are testing for catastrophic forgetting, you should: a) show scores for the unadapted model for comparison, and b) evaluate on a domain seen by the original/unadapted model.

---

> ### Author Response · Authors · 2024-11-20
> **Rebuttal by Authors (1/4)**
>
> We are grateful for the time and effort you have invested in reviewing our manuscript. We take each of your concerns seriously and are confident that we can address all issues raised to your satisfaction.
>
> ---
>
> **Weakness 1:**
>
> > This paper would benefit from clearly defining and justifying the task at hand. It addresses domain adaptation with target-side monolingual data, and that is well justified. However, there are several implicit assumptions throughout that do not seem to be stated clearly or given a justification. For example, a) assuming that they are starting from a generic trained model to adapt, and cannot train a model to the relevant domain from scratch (section 4.2), b) assuming that the adapted model shouldn't have catastrophic forgetting of the original domain (line 45), c) assuming that it is desirable to do well on tasks that were unseen in both the original model and the adaptation data (table 6), and d) that the adapted output should exhibit minimal changes compared to the original model (line 410). There is nothing wrong with these assumptions necessarily, but you need to clearly state them and justify why you are restricting your exploration to these.
>
> Thank you for your valuable comments.
>
> (a): We explained in the first paragraph of the introduction that obtaining parallel data for training on domain-specific data to create high-quality translation models is challenging.
>
> (b)-(d): We added supplementary information about the meaning of catastrophic forgetting. Specifically, training with limited data often leads to local optima and involves exploring a loss landscape that differs from that of pretraining. This results in parameters that diverge from the original optimal solution, causing performance degradation. Therefore, sufficient data is required for domain shifts.  In our main related work, Korbak et al., (2022), reinforcement learning is used for domain shifts. However, methods requiring scoring, like reinforcement learning, focus on task-specific learning, which can lead to a decline in generalization performance. Here, generalization performance refers to fundamental forgetting, such as a loss of fluency in generated sentences, resulting in unnatural text. Based on these observations, our approach is motivated by the goal of not only producing fluent translations but also ensuring consistency in domain-specific terminology to achieve effective domain shifts.
>
> Nevertheless, in response to this comment and Question 5, which related to the same problem, we conducted additional experiments to evaluate performance changes in the general domain and updated Table 6 in the manuscript.
>
> We attached the simple table as follows:
>
> |           |       | Conf. |  |  |  |  | BLEU | | | | |
> |-----------|-------|-----|------|-----|-------|-------|-----|------|-----|-------|-------|
> |           |       | Edu | Thes | Sci | G.f.t | G.d.c | Edu | Thes | Sci | G.f.t | G.d.c |
> | → zh      | Edu   | **7.94** | 8.29 | 9.21 | -1.16 | 8.52 | **1.09** | 0.29 | -0.44 | -0.76 | -0.17 |
> |           | Thes  | 6.70 | **3.99** | 5.58 | -0.31 | 7.69 | 0.87 | **0.20** | 0.37 | -0.28 | 0.13  |
> |           | Sci   | 4.83 | 4.20 | **5.38** | -0.68 | 4.92 | 0.87 | 0.50 | **0.28** | -0.02 | 0.33  |
> | → en      | Edu   | **7.39** | 7.86 | 7.83 | -0.59 | 8.31 | **0.69** | -0.07 | -0.27 | -0.11 | 0.19  |
> |           | Thes  | 7.72 | **8.46** | 8.03 | -0.51 | 8.84 | 0.66 | **-0.11** | 0.09 | -0.04 | 0.20  |
> |           | Sci   | 7.81 | 8.51 | **8.07** | -0.55 | 8.89 | 0.64 | -0.26 | **-0.02** | -0.07 | 0.26  |
> | → de      | IT    | **10.61** | 8.97 | 9.90 | -0.22 | 12.75 | **2.86** | 0.38 | -1.13 | -0.15 | -1.65 |
> |           | Med   | 8.32 | **6.61** | 7.11 | -0.27 | 9.47 | 2.44 | **0.28** | -0.63 | -0.07 | -0.90 |
> |           | Koran | 0.65 | 0.47 | **-0.09** | -0.21 | -0.86 | 1.05 | 0.93 | **-0.01** | -0.18 | -0.08 |
> | → en      | IT    | **5.89** | 6.17 | 6.76 | -0.22 | 10.38 | **2.72** | -0.78 | -0.04 | -0.11 | -0.81 |
> |           | Med   | -1.02 | **-0.05** | -0.82 | -0.29 | -1.50 | -0.65 | **-0.42** | -0.11 | -0.06 | -0.18 |
> |           | Koran | 6.07 | 5.92 | **5.95** | -0.20 | 8.28 | 0.97 | -0.91 | **0.13** | -0.14 | -0.40 |
>
> Specifically, we added the difference between Fine-tuned and Pre-trained and the difference between DCDPG and Pre-trained on a generic domain in Table 6 as a pivot for the relative difference between Fine-tuned and DCDPG in crossing domains. In this way, we can show the collapse of the vanilla fine-tuning methods, and the generalization of CDPG. Thank you for your suggestion!

---

> > ### Comment · Reviewer_n3SL · 2024-11-22
> > **Response to rebuttal 1/4**
> >
> > Thank you for your response!
> >
> > ```
> > 1a) "We explained in the first paragraph of the introduction that obtaining parallel data for training on domain-specific data to create high-quality translation models is challenging."
> > ```
> >
> > I don't think this response addresses my original statement ("assuming that they are starting from a generic trained model to adapt, and cannot train a model to the relevant domain from scratch"), so I'm sorry that I was unclear. It is possible to train a model from scratch on the (potentially small) in-domain parallel data *plus* the generic data. I did not mean to exclusively suggest training on only in-domain data.
> >
> > ```
> > (b)-(d): We added supplementary information about the meaning of catastrophic forgetting. Specifically, training with limited data often leads to local optima and involves exploring a loss landscape that differs from that of pretraining. This results in parameters that diverge from the original optimal solution, causing performance degradation. Therefore, sufficient data is required for domain shifts. In our main related work, Korbak et al., (2022), reinforcement learning is used for domain shifts. However, methods requiring scoring, like reinforcement learning, focus on task-specific learning, which can lead to a decline in generalization performance. Here, generalization performance refers to fundamental forgetting, such as a loss of fluency in generated sentences, resulting in unnatural text. Based on these observations, our approach is motivated by the goal of not only producing fluent translations but also ensuring consistency in domain-specific terminology to achieve effective domain shifts.
> > ```
> >
> > I apologize again -- I don't see how this response addresses my concerns. I think this is because I didn't express them clearly, so I will rephrase here:
> >
> > It is unclear to me why certain assumptions were made, and whether they would be widely applicable in a real-world setting. It is fine to make assumptions on the problem space addressed, but these should be spelled out clearly. Some assumptions that I felt were either not stated clearly enough or not justified well enough for real-world applicability were:
> >
> > 1. That the model needs to retain performance on the original domain (vs. simply doing well on the new domain)
> >
> > 2. That the model additionally needs to do well on domains that were not included in the original model or the new domain
> >
> > 3. That the translations outputted by the new model should be as similar as possible to the translations outputted by the new model.
> >
> > These properties may be desirable in some applications, but they are not universally desirable, so it should be justified why they are emphasized vs. other trade-offs.
> >
> > By the way, as also pointed out also by reviewer `UmaA`, I don't believe the definition of catastrophic forgetting being used in the paper is a generally accepted one, at least in MT domain adaptation, so it would be good to cite where it comes from.

---

> ### Author Response · Authors · 2024-11-20
> **Rebuttal by Authors (2/4)**
>
> **Weakness 2:**
>
>  > There are some issues with the evaluations. First, the base model that is used is OPUS-MT, but the domain-specific datasets that are used for evaluation for EN<->DE come from OPUS, so they were used to train the base model. Thus, this is not a true scenario of domain adaptation to an unseen domain, but one of domain shift. It is not clear to me whether this was done intentionally, but I think it would be preferable to do some domain adaptation evaluations with unseen data (and this might explain the lack of consistent positive results for any of the domain adaptation models, including the baselines and the EBM approach). Second, confidence is used as an evaluation score, when it is not clear that this correlates with any sort of meaningful MT evaluation. Third, the examples given in table 5 point more towards overfitting to the vocabulary of a specific dataset (not a specific domain) than to any true translation quality improvements.
>
>
> Thank you for your feedback.
>
> **First**, indeed, as pointed out, Aharoni et al. (2020)  created the dataset by automatic mining and multiple cleaning of OPUS, so the en <-> de data may be included in the OPUS dataset. However, we would like to emphasize the following: 1) Domain data constitutes only a small part of the training data. The model we use is generally trained on broad data, and domain adaptation serves to awaken its capability in a specific domain. 2) We manually refined the test set to ensure accurate evaluation. 3) We also considered en <-> zh data, where the Chinese dataset, UM-Corpus, is an indirect access dataset. For these reasons, we have already conducted experiments on unseen data, and the comprehensive evaluation results demonstrate that our method surpasses standard domain adaptation techniques.
>
> **Secondly**, a model with high confidence indicates that it can generate outputs with greater certainty relative to the domain distribution. For example, even if the scores before and after applying our method are tied, an increase in confidence implies that the model is better specialized for the specific domain. Additionally, we evaluated our method using multiple evaluation metrics, not just confidence. It is crucial to consider both confidence and MT metrics when analyzing the results. Confidence serves as an alternative angle of evaluation to reinforce MT metrics. Because these are independent evaluations, there is no requirement for correlation. In fact, their independence allows for a more multifaceted analysis.
>
> **Regarding the third point**, we are unsure how you would propose distinguishing between true domain adaptation and overfitting in your consideration. However, our method leverages the target-side word distribution for domain adaptation. As the number of target domain data increases, the domain distribution approaches the true distribution, enabling more accurate domain shifts. Furthermore, the examples we provided do not include words explicitly present in the constructed target domain distribution, indicating inductive generation. A detailed qualitative analysis of such unseen terminology is provided in Appendix I. This supports our claim that specific domain adaptation has been achieved. If you have any suggestions for better ways to present this, we would be grateful for your comments.
>
> ---
>
> **Weakness 3:**
>
> > This paper should take a broader view of the literature, including terminology-constrained machine translation (which seems to be hinted at as the ultimate goal of the proposed approach, e.g. in line 410 and table 5) as well as cases where the assumptions I listed in item 1 are relaxed (e.g., training a domain-specific model from scratch). In addition, the following paper is directly related (even without taking a broader view) and should be used as a baseline.
>
> We added literature on terminology-constrained machine translation and your recommendation of the paper to reference. However, the domain discussed in the recommended literature involves a large amount of data, which is not well-suited for the stress setting we assume in this study, where only a small amount of target domain data is used. Nonetheless, the experiments in the multi-domain setting presented in the paper are valuable references, and we have incorporated similar experiments into our study.
>
> Specifically, we added Appendix G and Table 9, which investigate the variations of Fine-tuned and CDPG in mixing two domains. The result shows that the vanilla fine-tuning methods are influenced by noisy data, because the performance decreases with the increase of noisy data. But, our proposed method, CDPG, shows more robust performance.

---

> > ### Author Response · Authors · 2024-11-20
> > **Rebuttal by Authors (3/4)**
> >
> > **Weakness 4:**
> >
> > > Beyond the missing citations, the baselines are insufficient or problematic. a) The proposed approach should be compared against LLM translation, both generic and using in-context learning with monolingual examples, the latter of which would directly address the problem at hand. b) The fine-tuning comparisons only evaluate i) fine-tuning the entire model and ii) fine-tuning only the attention weights. To me given the small dataset and focus on target-side data it would make sense to explore other approaches like fine-tuning the decoder only. c) Line 199 says "the checkpoint, which has the best performance on the development set, is measured for comparison." but line 193 says fine-tuning is done on the development set. So the models are fine-tuned on the same set that is used for checkpoint selection; it would not be surprising if they don't generalize well to the test set.
> >
> > To demonstrate the effectiveness of our training method, we conducted evaluations using the same model and the same data.
> >
> > (a) While it is true that LLMs are popular nowadays, using LLMs is not ideal as a baseline for demonstrating the effectiveness of our method because their training data differs significantly. Moreover, this paper focuses on proving the functionality of our method, and larger parameter models are simply one variation. We have noted this as a Limitation and also discussed it in the future directions.
> >
> > (b) As you are aware, our tuning settings are among the most standard ones. If we were to examine each part of the Transformer, such as the encoder, FFN, or specific layers, it is possible to explore various alternatives. However, exploring all these configurations goes beyond the scope of this study. We are researching a novel training method, not engaging in a SOTA competition or a comprehensive meta-evaluation.
> >
> > (c) Using test data to select checkpoints constitutes p-hacking. Thus, we argue that selecting checkpoints based on validation data is appropriate. Furthermore, since word distribution acquisition and actual translation evaluation are separate aspects of the data usage, your concern does not apply, and we believe our approach sufficiently generalizes.
> >
> > If you have additional concerns, please feel free to ask us!
> >
> > ---
> >
> > **Question 1:**
> >
> > > In line 36, you say "automatically collecting a sufficient amount of domain-specific parallel data is challenging". It would be good to get some quantitative information to justify this statement, particularly what you mean by "sufficient". In general, fine-tuning and ICL can work well with an extremely small corpus.
> >
> > Thank you. Generally, it is extremely challenging to obtain a sufficient amount of parallel domain-specific data, especially for specific domains such as corporate, organizational, internal documents, and personal data. While there are datasets available for a limited number of domains, the variety of domains is essentially infinite. While it is difficult to provide more than an intuitive explanation, such data is not readily available. Of course, using resources like LLMs can be an effective approach; however, such resources are not always accessible. Therefore, the approach we are taking with MT models remains valid. Applying our method to LLMs is a topic for future work. For now, we ask that you acknowledge the fact that our training method has proven effective in the stress-setting environment we envisioned.
> >
> > ---
> >
> > **Question 2:**
> >
> > > Line 44 says: "However, naively performing fine-tuning [...] can lead to catastrophic forgetting issues, such as the loss of fluency in the translated sentences acquired during pre-training, thereby causing a reduction in translation performance". Can you share evidence of this? Typically, catastrophic forgetting doesn't cause a loss of fluency in NMT per se, but just poorer performance on seen domains.
> >
> > It causes overfitting, resulting in high performance in specific domains but potentially leading to a loss of fluency in other domains. Additionally, fine-tuning methods using reinforcement learning approaches are prone to significant overfitting, which can cause a substantial decline in fluency (Korbak et al., 2022). There are varying levels of catastrophic forgetting, ranging from severe forgetting that impairs fluency to more moderate forms, such as domain adaptation, which only results in performance degradation in the original domain.
> >
> > However, from your good comments, we evaluated the performance changes in the general domain and updated Table 6 in the manuscript. Specifically, we added the difference between Fine-tuned and Pre-trained and the difference between DCDPG and Pre-trained on a generic domain in Table 6 as a pivot for the relative difference between Fine-tuned and DCDPG in crossing domains. In this way, we can show the collapse of the vanilla fine-tuning methods, and the generalization of CDPG.

---

> > > ### Author Response · Authors · 2024-11-20
> > > **Rebuttal by Authors (4/4)**
> > >
> > > **Question 3:**
> > >
> > > > It would be good to add a discussion of whether the EBMs increase the parameter size, memory footprint, or inference speed of the model.
> > >
> > > As you know, this method is purely a training approach and does not affect model size, memory usage, or inference speed, and we have included the statement in Section 2.
> > >
> > > ---
> > >
> > > **Question 4:**
> > >
> > > > Line 194 should cite the back-translation paper. Also, I would recommend including a comparison to back-translation as a baseline, and labeling "fine-tuned" as your upper bound.
> > >
> > > Thank you. Based on your suggestion, we added back-translation as a baseline in Appendix G. This highlights even more clearly that fine-tuning with clean data serves as the upper bound for back-translation.
> > >
> > > ---
> > >
> > > **Question 5:**
> > >
> > > > I found the presentation of table 6 extremely confusing. It would be clearer to simply show the BLEU scores of the two models, rather than showing the difference between them. In addition, if you are testing for catastrophic forgetting, you should: a) show scores for the unadapted model for comparison, and b) evaluate on a domain seen by the original/unadapted model.
> > >
> > > Thank you. Based on your feedback, we updated Table 6 as we mentioned in response to Weakness 1. Originally, we designed Table 6 to show the stronger generalization, therefore, we compared vanilla methods and our proposed methods. However, in order to resolve your question, we added the comparison between tuned models and pre-trained models on a generic domain to show the basic case in crossing domains. Based on this update, we not only can show the ability of generalization, but also can show the degradation in vanilla fine-tuning methods.
> > >
> > > ---
> > >
> > > We believe we have addressed all of your concerns, but if you have any further concerns, please do not hesitate to let us know! We will do our best to respond sincerely.

---

> > > ### Comment · Reviewer_n3SL · 2024-11-22
> > > **Response to rebuttal 3/4**
> > >
> > > 4a
> > > ```
> > > To demonstrate the effectiveness of our training method, we conducted evaluations using the same model and the same data.
> > > (a) While it is true that LLMs are popular nowadays, using LLMs is not ideal as a baseline for demonstrating the effectiveness of our method because their training data differs significantly. Moreover, this paper focuses on proving the functionality of our method, and larger parameter models are simply one variation. We have noted this as a Limitation and also discussed it in the future directions.
> > > ```
> > >
> > > Given the ubiquity and availability of LLMs, if they are able to do this task more effectively with the same in-domain data (regardless of the initial training data), then it would be good to clarify in what cases your method would be useful. For this reason, it would be good to compare to an LLM baseline.
> > >
> > > 4b
> > > ```
> > > (b) As you are aware, our tuning settings are among the most standard ones. If we were to examine each part of the Transformer, such as the encoder, FFN, or specific layers, it is possible to explore various alternatives. However, exploring all these configurations goes beyond the scope of this study. We are researching a novel training method, not engaging in a SOTA competition or a comprehensive meta-evaluation.
> > > ```
> > >
> > > Do you have a citation for fine-tuning *only the attention weights* being "among the most standard [tuning settings]"? Also, I'm sorry my writing was unclear; but I don't believe I suggested "examin[ing] each part of the Transformer, such as the encoder, FFN, or specific layers" or "engaging in a SOTA competition or a comprehensive meta-evaluation". I stand by my statement that given the small dataset and focus on target-side data, fine-tuning the decoder only is a more intuitive comparison than fine-tuning the attention weights only.
> > >
> > > 4c
> > > ```
> > > (c) Using test data to select checkpoints constitutes p-hacking. Thus, we argue that selecting checkpoints based on validation data is appropriate. Furthermore, since word distribution acquisition and actual translation evaluation are separate aspects of the data usage, your concern does not apply, and we believe our approach sufficiently generalizes.
> > > ```
> > > I am not sure what is the cause of the misunderstanding here. The original review points out that both checkpoint selection and fine-tuning are done on the same dataset; **nowhere** does it suggest "using test data to select checkpoints". Standard practice would be to use **separate** datasets for validation/checkpoint selection (development set), fine-tuning (typically called the training set), and evaluation (test set and held-out set). It seems (please correct me if I misread the lines cited in the original review) that you are using the same set for validation/checkpoint selection and fine-tuning, and a separate second set for evaluation.

---

> > > > ### Author Response · Authors · 2024-11-25
> > > > **Re: Rebuttal by Authors (1/3)**
> > > >
> > > > Thank you for your prompt response and detailed feedback! Based on your feedback, we have revised the manuscript again to clarify the motivation. The text highlighted in green indicates the areas that were added or modified in response to your comments.
> > > >
> > > > ---
> > > >
> > > > **1 (a)**
> > > >
> > > > >  I don't think this response addresses my original statement ("assuming that they are starting from a generic trained model to adapt, and cannot train a model to the relevant domain from scratch"), so I'm sorry that I was unclear. It is possible to train a model from scratch on the (potentially small) in-domain parallel data plus the generic data. I did not mean to exclusively suggest training on only in-domain data.
> > > >
> > > > Thank you, we understand your point. Indeed, among the various attempts at domain adaptation, we specifically focused on utilizing existing pre-trained general NMT models. This point is highlighted in green text in paragraphs 1 and 2 of Section 1: Introduction. By writing it this way, we believe it has become clearer why topics such as catastrophic forgetting are addressed in our study. Thank you for your valuable suggestion!
> > > >
> > > > > “However, when we shift the focus from parallel data to monolingual data, it is possible to easily obtain such monolingual data for the target domain, **and numerous pre-trained general NMT models have been developed**. In this study, **we focus on leveraging pre-trained general NMT models that are easily accessible and** attempt to transfer an NMT model pre-trained on a general domain into a domain-specific NMT model by using only the features obtained from the monolingual domain data of the translation target language.”
> > > >
> > > > ---
> > > >
> > > > **1 (b)-(d):**
> > > >
> > > > > 1 (b): That the model needs to retain performance on the original domain (vs. simply doing well on the new domain)
> > > >
> > > > > By the way, as also pointed out also by reviewer UmaA, I don't believe the definition of catastrophic forgetting being used in the paper is a generally accepted one, at least in MT domain adaptation, so it would be good to cite where it comes from.
> > > >
> > > > Thank you for your comments. Based on your comments and the additional experiments conducted during the rebuttal period, we have defined catastrophic forgetting as “ranging from the loss of fluency in translated sentences acquired during pre-training to degradation in non-specific domains caused by overfitting to specific terminologies, thereby causing a reduction in translation performance” in the introduction section. We believe this clarification has made the meaning of catastrophic forgetting more explicit. Additionally, thank you for your cooperation in refining the manuscript during the rebuttal period. Including this phrasing has indeed made the text more comprehensible.
> > > >
> > > > > **1 (c)**: That the model additionally needs to do well on domains that were not included in the original model or the new domain
> > > >
> > > > We did not state that adaptation to unseen domains is desirable. Instead, the results demonstrate that the model fits the target domain without catastrophic forgetting. This presentation style is standard practice [1].
> > > >
> > > > [1]: Unsupervised Domain Clusters in Pretrained Language Models (Aharoni & Goldberg, ACL 2020)
> > > >
> > > > ---
> > > >
> > > > > **1 (d):** That the translations outputted by the new model should be as similar as possible to the translations outputted by the new model.
> > > >
> > > > (Perhaps it is a typo from "old model"), as you mentioned, one of CDPG’s key advantages lies in its ability to approach the target distribution while maintaining the original distribution as much as possible. Its objective, summarized as “harmlessly modifying the model’s knowledge to avoid degrading generalization performance or excessive overfitting to a specific domain,” has now been added to the introduction. Thank you for pointing this out!

---

> > > > > ### Author Response · Authors · 2024-11-25
> > > > > **Re: Rebuttal by Authors (2/3)**
> > > > >
> > > > > > **2.1.1:** I believe that the fact that the test set is likely to be contained in the training set for en<->de makes the evaluation invalid. Can you please clarify why "manually refin[ing] the test set to ensure accurate evaluation" remedies this issue?
> > > > >
> > > > > > **2.1.2:** Thank you for including en<->zh on unseen data. Regardless, the paper should include only these experiments and not en<->de since the training data likely includes the test data for en<->de.
> > > > >
> > > > >
> > > > > Thank you for your feedback. It is well known that neural network models, such as language models and NMT models, tend to memorize training data [2]. To address this, we manually refined the en<->de test set, ensuring it slightly differs from the original sentences. As a result, the refined test set becomes unseen data where memorized knowledge cannot be utilized, and we believe it is largely unaffected by contamination [3]. We also plan to release this refined test set after acceptance. Consequently, we consider the evaluation on en<->de valid and have included the en<->de data in the table.
> > > > >
> > > > > [2] Memorisation Cartography: Mapping out the Memorisation-Generalisation Continuum in Neural Machine Translation (Dankers et al., EMNLP 2023)
> > > > >
> > > > > [3]: Finding Memo: Extractive Memorization in Constrained Sequence Generation Tasks (Raunak & Menezes, EMNLP Findings 2022)
> > > > >
> > > > > ---
> > > > >
> > > > > > **2.2** Can you share citations you have for these assertions, or for confidence being a good metric for MT evaluation? Can you share evidence that your models are well-calibrated? Similar to 2.1, inclusion of valid experiments or metrics does not justify the inclusion of invalid experiments or metrics.
> > > > >
> > > > > Confidence represents the probability associated with the output generated by a neural network [4, 5]. Higher confidence reflects greater certainty in the output, while lower confidence indicates higher uncertainty. Ideally, models should generate correct outputs with high confidence. This approach has been applied in NMT [6, 7], supporting its validity as one of the evaluation aspects. Additionally, we assessed our method using multiple evaluation metrics beyond confidence. A comprehensive analysis requires jointly considering confidence and MT metrics. We added references to the manuscript.
> > > > >
> > > > > [4]: On Calibration of Modern Neural Networks (Guo et al., ICML 2017)
> > > > >
> > > > > [5]: Revisiting the Calibration of Modern Neural Networks (Minderer et al., NeurIPS 2021)
> > > > >
> > > > > [6]: When Does Label Smoothing Help? (Müller et al., NeurIPS 2019)
> > > > >
> > > > > [7]: On the Inference Calibration of Neural Machine Translation (Wang et al., ACL 2020)
> > > > >
> > > > > ---
> > > > >
> > > > > > **2.3:** Thanks, I took a look at Appendix I as well. I am not sure that Case 2 in appendix I is relevant, as "Tunnelgeräts" and "Tunnelgerätes" are both correct variations of the same word. By "the examples we provided do not include words explicitly present in the constructed target domain distribution, indicating inductive generation", do you mean that something like "Tunnelgerätes" (and subwords thereof) does not occur in the in-domain target data? That's interesting behavior, then!
> > > > >
> > > > > I’m glad we could spark your interest! As an additional experiment, we identified an example: “Tunnelgerätes.” Despite not being present in the target domain data, our method successfully generated this term. This demonstrates the strength of our approach compared to terminology-constrained methods mentioned in **your comment 3 about related works**. We believe this finding reinforces the validity of our method. If we discover better examples, we will incorporate them in the camera-ready version.

---

> > ### Comment · Reviewer_n3SL · 2024-11-22
> > **Response to rebuttal 2/4**
> >
> > 2.1.
> > ```
> > First, indeed, as pointed out, Aharoni et al. (2020) created the dataset by automatic mining and multiple cleaning of OPUS, so the en <-> de data may be included in the OPUS dataset. However, we would like to emphasize the following: 1) Domain data constitutes only a small part of the training data. The model we use is generally trained on broad data, and domain adaptation serves to awaken its capability in a specific domain. 2) We manually refined the test set to ensure accurate evaluation. 3) We also considered en <-> zh data, where the Chinese dataset, UM-Corpus, is an indirect access dataset. For these reasons, we have already conducted experiments on unseen data, and the comprehensive evaluation results demonstrate that our method surpasses standard domain adaptation techniques.
> > ```
> >
> > 1. I believe that the fact that the test set is likely to be contained in the training set for en<->de makes the evaluation invalid. Can you please clarify why "manually refin[ing] the test set to ensure accurate evaluation" remedies this issue?
> >
> > 2. Thank you for including en<->zh on unseen data. Regardless, the paper should include only these experiments and not en<->de since the training data likely includes the test data for en<->de.
> >
> > 2.2
> > ```
> > Secondly, a model with high confidence indicates that it can generate outputs with greater certainty relative to the domain distribution. For example, even if the scores before and after applying our method are tied, an increase in confidence implies that the model is better specialized for the specific domain. Additionally, we evaluated our method using multiple evaluation metrics, not just confidence. It is crucial to consider both confidence and MT metrics when analyzing the results. Confidence serves as an alternative angle of evaluation to reinforce MT metrics. Because these are independent evaluations, there is no requirement for correlation. In fact, their independence allows for a more multifaceted analysis.
> > ```
> > Can you share citations you have for these assertions, or for confidence being a good metric for MT evaluation? Can you share evidence that your models are well-calibrated? Similar to 2.1, inclusion of valid experiments or metrics does not justify the inclusion of invalid experiments or metrics.
> >
> > 2.3
> > ```
> > Regarding the third point, we are unsure how you would propose distinguishing between true domain adaptation and overfitting in your consideration. However, our method leverages the target-side word distribution for domain adaptation. As the number of target domain data increases, the domain distribution approaches the true distribution, enabling more accurate domain shifts. Furthermore, the examples we provided do not include words explicitly present in the constructed target domain distribution, indicating inductive generation. A detailed qualitative analysis of such unseen terminology is provided in Appendix I. This supports our claim that specific domain adaptation has been achieved. If you have any suggestions for better ways to present this, we would be grateful for your comments.
> > ```
> > Thanks, I took a look at Appendix I as well. I am not sure that Case 2 in appendix I is relevant, as "Tunnelgeräts" and "Tunnelgerätes" are both correct variations of the same word.
> > By "the examples we provided do not include words explicitly present in the constructed target domain distribution, indicating inductive generation", do you mean that something like "Tunnelgerätes" (and subwords thereof) does not occur in the in-domain target data? That's interesting behavior, then!
> >
> > 3. Thank you for adding a discussion of terminology-constrained MT. I think given the assumptions made in the paper that this is the most similar setup to the one you are interested in.

---

> ### Author Response · Authors · 2024-11-25
> **Re: Rebuttal by Authors (3/3)**
>
> > **4.a:** Given the ubiquity and availability of LLMs, if they are able to do this task more effectively with the same in-domain data (regardless of the initial training data), then it would be good to clarify in what cases your method would be useful. For this reason, it would be good to compare to an LLM baseline.
>
> Thank you. To clarify our understanding, when you refer to LLMs, are you specifically referring to decoder-side language models like LLaMA, rather than pre-trained models in general? While your suggestion is intriguing, our study focuses on encoder-decoder models, and we consider discussions about decoder-only models to be out of scope.
>
> As you correctly pointed out, our research emphasizes the “**ubiquity and availability of pre-trained NMT models”**, which we highlighted in response to your comment 1(a) by explicitly stating this in the introduction. We believe this addition has clarified the scope of our study.
>
> Given the focus on “**ubiquity and availability of pre-trained NMT models”**, we argue that our experiments are appropriately designed to be comparable and sufficiently demonstrate the practical utility of our method in addressing “**what cases our method would be useful”**.
>
> ---
>
> > **4.b:** Do you have a citation for fine-tuning only the attention weights being "among the most standard [tuning settings]"? Also, I'm sorry my writing was unclear; but I don't believe I suggested "examin[ing] each part of the Transformer, such as the encoder, FFN, or specific layers" or "engaging in a SOTA competition or a comprehensive meta-evaluation". I stand by my statement that given the small dataset and focus on target-side data, fine-tuning the decoder only is a more intuitive comparison than fine-tuning the attention weights only.
>
> Thank you for your explanation. Do you have any references to support your opinion? In our study, we first conducted full fine-tuning for a fair comparison with CDPG tuning that. Next, we used LoRA to update only the attention weights, following the standard setting (Hu et al., 2021) as mentioned in Section 4.2.
>
> Our paper primarily focuses on introducing a new tuning strategy, and we believe it is appropriate to compare results under the same tuning conditions. For a fair comparison, we assert that tuning both the encoder and decoder parameters is the most reasonable setup.
>
> ---
>
> > **4.c:** I am not sure what is the cause of the misunderstanding here. The original review points out that both checkpoint selection and fine-tuning are done on the same dataset; nowhere does it suggest "using test data to select checkpoints". Standard practice would be to use separate datasets for validation/checkpoint selection (development set), fine-tuning (typically called the training set), and evaluation (test set and held-out set). It seems (please correct me if I misread the lines cited in the original review) that you are using the same set for validation/checkpoint selection and fine-tuning, and a separate second set for evaluation.
>
> Thank you for clarifying your question. First, the "features" used for CDPG are derived from the validation set, but since the validation set data is not directly used during training, this does not affect the checkpoint selection for CDPG. Furthermore, to ensure fairness, we also use the validation set for training the Fine-tuned (i.e., vanilla fine-tuning) and LoRA. Given the scarcity of domain-specific data (2,000 samples for German and 3,000 samples for Chinese), further splitting the data would reduce the effectiveness of training (small validation set is not effective; big validation set harms the training).
>
> Considering that our setup involves fine-tuning with a small learning rate over 10 epochs, the risk of overfitting is relatively low. Therefore, we select checkpoints based on their inference performance (i.e., BLEU scores and BERTScores) on the validation set, rather than relying on the model’s loss on the validation set (equivalent to the training loss). Empirically, the selected checkpoints are not always the last ones, but even when the last checkpoint is chosen, it consistently brings improvements in testing within the respective domain compared to the pre-trained models.
>
> ---
>
> Thank you for your effort in reviewing our manuscript again. We believe we have addressed all of your concerns. However, if you have any further questions or additional concerns, please don’t hesitate to share them. We are committed to responding sincerely and thoroughly within the remaining time.
>
> We look forward to your reply.

---

> > ### Author Response · Authors · 2024-12-02
> >
> > Dear Reviewer n3SL,
> >
> > Thank you for your time and effort in reviewing our manuscript and engaging in the discussion process. As the discussion period nears its conclusion, we hope that our responses and revisions have effectively addressed your concerns.
> >
> > If there are any remaining questions or concerns, please let us know in detail at your earliest convenience. We will do our best to address them promptly before the discussion phase ends.
> >
> > If your concerns have been resolved, we kindly ask you to consider positively revising your evaluation to reflect the improvements in our work.
> >
> > Thank you again for your valuable feedback and dedication.
> >
> > Best regards,
> >
> > Anonymous Authors

---

> ### Author Response · Authors · 2024-12-04
> **Thank you for participating in the author-reviewer discussion phase**
>
> Dear Reviewer n3SL,
>
> We understand that you may be busy, and we greatly appreciate the time and effort you have already dedicated to this review process. We believe that your lack of response to our rebuttals is not due to irresponsibly abandoning your role as a reviewer, but rather because the concerns regarding this paper have been resolved. Sorry to bother you, but if the concerns have indeed been addressed, we kindly request you to **update your scores** for **Soundness**, **Presentation**, **Contribution**, and **Overall Rating** to reflect the resolution of all issues, as is typically expected of reviewers following the discussion period.
>
> Best regards,
>
> Anonymous Authors

---

### Official Review · Reviewer_UmaA · 2024-10-31

**Soundness:** 2
**Presentation:** 2
**Contribution:** 2
**Rating:** 3
**Confidence:** 3

**Summary:**

This paper looks at modeling domain-specific translation by formulating the target domain as a conditional energy based model (EBM), and approximates the EBM with conditional distributional policy gradients (CDPG). The approximation is done by using unigram probabilities to model the target domain, create binary features of when unigrams appear in target sentences, and fine-tune the pre-trained translation model according to the constructed EBM. The authors also propose a dynamic variation of CDPG specific to autoregressive modeling. On en <-> zh and en <-> de models, the method is compared to the original model, a 3k sample fine-tuned model, and a LoRA fine-tuned model.

**Strengths:**

1. Energy-based models have been well established for controllable generation and using them in the context of domain-specific MT is a solid application in an important problem.
2. This method can apply on top of pre-existing models, as the authors show with the OPUS-MT models. This improves its extensibility where a model does not need to be trained from scratch to begin with.
3. The method outperforms a normal fine-tuning based approach for a reduced number of fine-tuning sentences, across 4 language directions.

**Weaknesses:**

1. My biggest concern with this work is its limited novelty especially compared to the work of Korbak et al. 2022. This work proposes how to represent the feature probabilities $\mu$ with unigram probabilities, but that seems to be all beyond the Korbak paper. Some discussion of exactly where this work differs would help differentiate them and highlight novelty.
2. The use of EBMs and CDPG is motivated by avoiding catastrophic forgetting, but there does not seem to be evidence of this in the results. In the intro, catastrophic forgetting is equated to reduction in translation performance, which is not quite what other work defines as catastrophic forgetting. Catastrophic forgetting refers to becoming worse on the original data distribution - showing improved performance on the target domain is not enough to evaluate catastrophic forgetting. The methods should ideally be evaluated on the original domain as well in order to show this.
3. Calling this method “monolingual data only” or unsupervised does not seem to work well with proposing dynamic CDPG as a main contribution in this work. These descriptors seem to only target the extension of vanilla CDPG to MT. These descriptors seem to only target the extension of original CDPG to MT.
4. There is a lack of implementation details regarding the EBM models and the fine-tuning procedure.

**Questions:**

1. What is the significance test you are running in the main tables?
2. Why use BERTScore over COMET? The latter is generally better for evaluating translations.
3. What is needed for the implementation of the EBMs? Some pseudocode/algorithm/code outlining the creation of the EBMs and the fine-tuning of the translation models would be helpful.

---

> ### Author Response · Authors · 2024-11-20
> **Rebuttal by Authors (1/2)**
>
> Thank you very much for your insightful comments. We are encouraged by your comments.
>
> ---
>
> **Weakness 1:**
> > My biggest concern with this work is its limited novelty especially compared to the work of Korbak et al. 2022. This work proposes how to represent the feature probabilities μ with unigram probabilities, but that seems to be all beyond the Korbak paper. Some discussion of exactly where this work differs would help differentiate them and highlight novelty.
>
> Thank you for your feedback. Regarding the differences from Korbak et al., we have addressed this starting from line 51: “Korbak et al. (2022) had only verified the effectiveness of CDPG for small shifts, such as translating numeral nouns (e.g., ‘two’) as digits (e.g., ‘2’). We extend the framework by using the token-level statistics of the target domain as features and constructing a large number of EBMs, approximating these to meet their constraints. Specifically, we shift the pre-trained NMT models toward the token-level unigram distribution of the target domain by CDPG, enabling domain shifts that better consider the frequency information of the entire target domain.” We believe this explanation clarifies the differences between our approach and theirs.
>
> ---
>
> **Weakness 2:**
>
> > The use of EBMs and CDPG is motivated by avoiding catastrophic forgetting, but there does not seem to be evidence of this in the results. In the intro, catastrophic forgetting is equated to reduction in translation performance, which is not quite what other work defines as catastrophic forgetting. Catastrophic forgetting refers to becoming worse on the original data distribution - showing improved performance on the target domain is not enough to evaluate catastrophic forgetting. The methods should ideally be evaluated on the original domain as well in order to show this.
>
> Thank you. Some studies on domain adaptation for NMT consider the results after a domain shift. Notably, Korbak et al. (2022) examined how their approach enables stable control without causing catastrophic forgetting in the target domain compared to general reinforcement learning methods. One definition of catastrophic forgetting refers to the situation where, while achieving the desired control, models often lose the ability to generate fluent sentences that original models can, as described in the introduction’s second paragraph. We evaluated the performance changes in the general domain and updated Table 6 in the manuscript. Specifically, we added the difference between Fine-tuned and Pre-trained and the difference between DCDPG and Pre-trained on a generic domain in Table 6 as a pivot for the relative difference between Fine-tuned and DCDPG in crossing domains. The results show the collapse of the vanilla fine-tuning methods and the generalization capacity of CDPG.
>
> ---
>
> **Weakness 3:**
>
> > Calling this method “monolingual data only” or unsupervised does not seem to work well with proposing dynamic CDPG as a main contribution in this work. These descriptors seem to only target the extension of vanilla CDPG to MT. These descriptors seem to only target the extension of original CDPG to MT.
>
> CDPG is sensitive to changes in the top_p parameter, which affects its quality. To address this, we propose Dynamic CDPG as an auxiliary approach, attempting to optimize parameter settings using a small amount of bilingual data. As indicated by the title, our main contribution is CDPG, while Dynamic CDPG is presented as a variant to emphasize that even when using CDPG, comparable results can be achieved to those of Dynamic CDPG, which performs dynamic parameter exploration. Therefore, as you pointed out, our main contribution is vanilla CDPG, and Dynamic CDPG serves as a variant to highlight and complement vanilla CDPG.
>
> ---
>
> **Weakness 4:**
>
> > There is a lack of implementation details regarding the EBM models and the fine-tuning procedure.
>
> Thank you. We used the disco library released by Kruszewski et al. (2023) as the base for our implementation. The disco library supports methods such as Korbak et al. (2022) and provides basic apis such as EBM and trainer. When using a large number of EBMs, computational bottlenecks can arise, for example, during scoring. To address this, we optimized the code and incorporated techniques such as matrix operations to improve processing speed. Please note that these optimizations are purely technical and do not affect the mathematical formulations. We have included details about the libraries used in the Experimental Setup section.
>
> [1] disco: a toolkit for Distributional Control of Generative Models (Kruszewski et al., ACL 2023)

---

> > ### Author Response · Authors · 2024-11-20
> > **Rebuttal by Authors (2/2)**
> >
> > **Question 1:**
> >
> > > What is the significance test you are running in the main tables?
> >
> > Thank you for the question. As mentioned in the 4.3 Evaluation section, we conducted statistical testing (paired bootstrap resampling) (Kohen, 2004) to verify whether the improvements in scores were statistically significant. This is a standard validation method frequently used in machine translation research. Statistically significant differences demonstrate that our method achieves performance improvements reliably.
> >
> > [2] Statistical Significance Tests for Machine Translation Evaluation (Koehn, EMNLP 2004)
> >
> > ---
> >
> > **Question 2:**
> >
> > > Why use BERTScore over COMET? The latter is generally better for evaluating translations.
> >
> > BERTScore calculates recall and precision by considering the meaning at the token level, enabling more detailed evaluations compared to sentence-level metrics like COMET.
> >
> > Furthermore, footnote 12 in Section 4.3 Evaluation states that the neural fine-tuned metric COMET is trained using data from WMT evaluation tasks. As a result, while it shows high correlation with human judgments for in-domain data, it has been shown that metrics like BLEU or embedding-based BERTScore exhibit higher correlation for out-of-domain data, especially with respect to domain-specific data.
> >
> > For this reason, we prioritized BERTScore as our main evaluation metric over COMET and similar alternatives. Nevertheless, from your comment, we added COMET results as a sentence-level metric in Appendix G. These results not only highlight the challenges COMET faces with domain-specific data again but also partially support our claims.
> >
> >  Adding COMET has made the paper’s narrative clearer and more accessible to readers. Thank you for your feedback!
> >
> > ---
> >
> > **Question 3:**
> >
> > > What is needed for the implementation of the EBMs? Some pseudocode/algorithm/code outlining the creation of the EBMs and the fine-tuning of the translation models would be helpful.
> >
> > As mentioned in Weakness 4, we use disco, a library that compiles EBM-based fine-tuning methods, including CDPG. Therefore, please defer their paper in our manuscript for details on how to use the library. While we have made slight modifications to improve computational speed, these changes do not alter the interface and primarily involve coding techniques to reduce computational costs. Since this manuscript is an academic paper rather than a technical report or demonstration paper, we decided not to include these implementation details. However, we plan to make the code publicly available after publication.
> >
> > ---
> > We believe we have addressed all of your concerns, but if you have any additional concerns, please don’t hesitate to let us know. We will do our best to respond sincerely within the given time constraints.

---

> > > ### Comment · Reviewer_UmaA · 2024-11-21
> > > **Response to rebuttal**
> > >
> > > Thanks for your detailed reply. I've enumerated each of the discussion points:
> > > Weaknesses responses:
> > > 1. **Differences from Korbak et al 2022:** Thanks for enumerating these differences in the PDF. While I see the number of EBMs is much larger, the contribution seems mostly related to details of its application which is still limited.
> > > 2. **Catastrophic forgetting**: Both fine-tuning and CDPG-methods seem to differ from the pre-trained evaluation similarly. I don't believe this shows enhanced generalization despite average token probability scores increasing.
> > > 3. **Monolingual data only**: I understand now to view DCDPG as a top-line after aprameter exploration. I am still concerned about the novelty of the main contribution, which is vanilla CDPG on NMT (CDPG from Korbak et al 2022)
> > > 4. **Implementation details**: Thanks for these details.
> > >
> > > Thanks for pointing out the significance test, and including COMET scores.
> > > I maintain my score due to the lack of support for one of the main claims of catastrophic forgetting, as well as the incremental nature of this work over the original CDPG work.

---

> > > > ### Author Response · Authors · 2024-11-25
> > > > **Re: Rebuttal by Authors**
> > > >
> > > > Thank you for your response! We are glad that all of the questions and the main claims in Weaknesses 3 and 4 have been resolved. We would like to further address the concerns that have not yet been resolved.
> > > >
> > > > ---
> > > >
> > > > > **1. Differences from Korbak et al 2022**
> > > >
> > > > Thank you for your valuable feedback. As you know, this study focuses on application research. Korbak et al. (2022) proposed the **CDPG training method** and demonstrated its potential by evaluating it with a small number of EBMs. However, their work was limited to artificial settings and did not explore actual downstream task applications.
> > > >
> > > > In practice, a straightforward implementation of CDPG faces scalability issues, such as the need to score every EBM, which results in significant computational and time costs, making it challenging to use a large number of EBMs as required for practical applications.
> > > >
> > > > This limitation explains why Korbak et al. (2022) and follow-up works have not attempted to apply CDPG to real downstream tasks. Please see the papers citing Korbak et al. (2022); their work primarily focuses on developing a new method.
> > > > In this study, we addressed these limitations by carefully optimizing the code to improve computational efficiency, enabling experiments with a large number of EBMs. This advancement made it possible, for the first time, to apply CDPG to a real-world domain adaptation task, demonstrating its practical utility. We will definitely release the code after acceptance.
> > > > To further illustrate our contribution, consider the leap from GPT-2 to GPT-3. While it might appear incremental (merely increasing parameters and training data), it unveiled "emergent abilities", significantly advancing real-world applications. Similarly, our study extends the potential of CDPG by scaling the number of EBMs and verifying its effectiveness through comprehensive analyses. This demonstrates the feasibility of applying CDPG to practical applications for the first time. While Korbak et al. (2022) proposed the method of CDPG, we applied it to real-world tasks. We argue that both **methodological contributions** and their **application contributions** are equally important.
> > > >
> > > > Furthermore, we submitted to “**applications to computer vision, audio, language, and other modalities**” area in ICLR, which specifically emphasizes applications. Considering this, we believe that focusing on applications aligns with one of the purposes of ICLR. While you may assess the merits of incremental work, we do not believe this justifies rejection, especially when **the area itself encourages application research**.
> > > >
> > > > We hope you will consider our contributions favorably. Thank you again for your thoughtful review and understanding.
> > > >
> > > > ---
> > > >
> > > > > **2. Catastrophic forgetting**
> > > >
> > > > Thank you for your further feedback.
> > > >
> > > > First, as mentioned in Section 4.2 and Appendix E, we fine-tuned both “Fine-tuned” and “LoRA” with a very small learning rate. This choice is due to our observation that fine-tuning on such small domain-specific datasets (2k–3k instances) tends to collapse easily. Our setup ensures that fine-tuning always improves performance in the corresponding domain.
> > > >
> > > > Consequently, because the updates are minimal, the forgetting effect in the generic domain is also limited. However, as shown in **Table 6**, we observe that fine-tuning **consistently leads to a decrease in both confidence and performance** in the generic domain.
> > > >
> > > > On the other hand, the significant changes in confidence indicate that CDPG introduces larger updates. In this case, the fact that CDPG does not always degrade performance in the generic domain demonstrates its stability. Notably, as discussed in Section 6.1 and Table 5, the changes introduced by CDPG are primarily token-level adjustments, which explains why CDPG maintains relative stability in performance on the generic domain despite significant fluctuations in confidence.
> > > >
> > > > From these results, we believe that when applying CDPG, we successfully achieve domain adaptation to the target domain while minimizing performance degradation in the generic domain compared to **Fine-tuned** results. This aligns with our objectives. Your suggestion to evaluate performance in the generic domain helped us validate this claim from both confidence and performance perspectives.
> > > >
> > > > It is worth noting that while mitigating catastrophic forgetting in the generic domain is important, our primary goal is to achieve effective domain adaptation to the target domain. The observation that CDPG achieves both objectives reinforces our claim.
> > > >
> > > > ---
> > > >
> > > > If you have further concerns or require additional clarification, please let us know. We are committed to addressing your concerns sincerely and comprehensively.
> > > >
> > > > Thank you for your time and consideration.
> > > >
> > > > We look forward to your reply.

---

> > > > > ### Author Response · Authors · 2024-12-02
> > > > >
> > > > > Dear Reviewer UmaA,
> > > > >
> > > > > Thank you for your time and effort in reviewing our manuscript and engaging in the discussion process. As the discussion period nears its conclusion, we hope that our responses and revisions have effectively addressed your concerns.
> > > > >
> > > > > If there are any remaining questions or concerns, please let us know in detail at your earliest convenience. We will do our best to address them promptly before the discussion phase ends.
> > > > >
> > > > > If your concerns have been resolved, we kindly ask you to consider positively revising your evaluation to reflect the improvements in our work.
> > > > >
> > > > > Thank you again for your valuable feedback and dedication.
> > > > >
> > > > > Best regards,
> > > > >
> > > > > Anonymous Authors

---

> ### Author Response · Authors · 2024-12-04
> **Thank you for participating in the author-reviewer discussion phase**
>
> Dear Reviewer UmaA,
>
> We understand that you may be busy, and we greatly appreciate the time and effort you have already dedicated to this review process. We believe that your lack of response to our rebuttals is not due to irresponsibly abandoning your role as a reviewer, but rather because the concerns regarding this paper have been resolved. Sorry to bother you, but if the concerns have indeed been addressed, we kindly request you to **update your scores** for **Soundness**, **Presentation**, **Contribution**, and **Overall Rating** to reflect the resolution of all issues, as is typically expected of reviewers following the discussion period.
>
> Best regards,
>
> Anonymous Authors

---

### Official Review · Reviewer_VUPi · 2024-11-05

**Soundness:** 3
**Presentation:** 3
**Contribution:** 2
**Rating:** 6
**Confidence:** 2

**Summary:**

This paper presents a new domain adaptation method for adapting a pretrained NMT model with low-resource monolingual in-domain data. The method employs conditional distribution policy gradients to approximate domain-specific features in the target languages, where the domain-specific features are represented by unigram distributions. The authors also propose dynamic CDPG, which dynamically adjusts parameters using a small bilingual validation sample. Experimental results show that their framework achieves improvements in some domains, primarily due to the model's enhanced learning of domain-specific words.

**Strengths:**

- Novel approach for domain adaptation using only monolingual data: The paper presents energy-based models to effectively adapt to monolingual domain-specific corpora.
- Extensive evaluation: The paper includes a variety of domains and languages to validate the proposed methods and provides additional analysis of improvements by assessing domain-specific terminology.

**Weaknesses:**

- Concerns with practicality and scalability: The setup in this paper raises concerns about practical applicability. In real-world scenarios, domain-specific corpora may often be more abundant, even when restricted to monolingual resources, and may differ significantly in size compared to the data setup in the paper. It remains unclear whether the proposed framework would can show scale well with larger datasets or whether it would overfit to small monolingual training sets, potentially only capturing domain features seen during training. This could lead to an over-reliance on vocabulary or terms specifically within the training set, rather than robust. The results may reflect high performance in producing domain-specific terms, but this may primarily apply to cases where test instances closely match the training domain vocabulary, potentially limiting broader generalization.

- Evaluation metric suitability: The selected evaluation metrics may not fully demonstrate the strengths of the proposed approach. The confidence scores of softmax indomain translation is somewhat concerning, as this does not necessarily correlate with successful adaptation. Moreover, author did not consider other naturalness metric  like fluency or advanced model based quality estimation metric, such as COMET. These metrics could provide a more reliable measure of adaptation performance and reveal whether the model truly maintains translation quality.

**Questions:**

- Domain feature generalization: One question is whether the model truly learns and generalizes domain features using this method. Specifically, I am curious if, during testing, the model can generate domain-specific terms that were not present in the monolingual training data but are consistent with the target domain's linguistic characteristics. If the model can accomplish this, it would indicate a deeper understanding of domain features rather than merely reproducing the training vocabulary. A more detailed analysis or experiments testing this capability would be valuable.

- Sensitivity to validation bilingual set in DCDPG: Given that DCDPG relies on a bilingual validation set for dynamic parameter tuning, I wonder if the method’s performance is sensitive to the quality and representativeness of this validation set. Could performance vary significantly depending on the domain alignment or size of the validation set used? This would be an important consideration for practitioners, as varying validation sets could lead to inconsistent results in practice.

---

> ### Author Response · Authors · 2024-11-20
> **Rebuttal by Authors**
>
> Thank you for your constructive feedback. Your thoughtful and detailed comments would be instrumental in strengthening our paper and clarifying our arguments.
>
> ---
>
> **Weakness 1: Concerns with practicality and scalability**
>
> Thank you for the interesting feedback. This study relies solely on the word distribution of the target domain. As the size of the data increases, the distribution approaches the true target domain, enabling higher-quality domain adaptation. At the same time, this study demonstrates that the approach works sufficiently well even with a small amount of available data like stress setting. Therefore, we believe it scales effectively. Furthermore, under the stress settings with limited domain data we aim at, the scalability issue is not critical.
>
> Additionally, we conducted experiments on robustness. By mixing multiple domains during training, we examined whether the method exhibits over-reliance. We added Appendix G and Table 9, which compare Fine-tuned and CDPG in mixing two domains. The result shows that the vanilla fine-tuning methods are influenced by noisy data, in which the performance decreases with the increase of noisy data. However, our proposed method, CDPG, shows more robust performance.
>
> Your feedback has helped us emphasize the merits of this paper even more clearly. Thank you!
>
> ---
>
> **Weakness 2: Evaluation metric suitability**
>
> BERTScore allows for more detailed evaluation compared to sentence-level metrics like COMET, as it considers token-level meaning and calculates recall and precision.
>
> We have added footnote 12 in Section 4.3 Evaluation, stating that the neural fine-tuned metric COMET is trained using data from WMT evaluation tasks. As a result, while it shows high correlation with human judgments for in-domain data, it has been shown that metrics like BLEU or embedding-based BERTScore exhibit higher correlation for out-of-domain data, especially with respect to domain-specific data.
>
> For this reason, we prioritized BERTScore as the main evaluation metric over COMET and similar metrics. Nevertheless, we included COMET as a sentence-level metric in Appendix G. These results not only highlight the challenges COMET faces with domain-specific data again but also partially support our claims. Incorporating COMET has made the paper’s narrative more accessible and easier for readers to follow. Thank you for your suggestion!
>
> ---
>
> **Question 1: Domain feature generalization**
>
> Thank you for pointing out this interesting question! We added Appendix I in the manuscript to show the deeper influence brought by CDPG from two instances.
> We have indicated in our manuscript that CDPG will increase the confidence of models. As an additional influence, the repeats in pre-trained models are resolved, like this instance:
>
> **Inference of Pre-trained:** *PPM. - Nein, nein, nein, nein, nein, nein, nein, nein, nein, nein…*
>
> **Inference of CDPG:** *PPM.*
>
> Then, we also can find some interesting instances, like this:
>
> **Inference of Pre-trained:** *Dies ist der Typ Ihres Tunnelgeräts.*
>
> **Inference of CDPG:** *Dies ist der Typ Ihres Tunnelgerätes.*
>
> Here, the “Tunnelgerätes” hits the reference and the term by fixing the original inaccurate word “Tunnelgeräts”. Meanwhile, “Tunnelgerätes” is not a feature used in fine-tuning! Therefore, this instance shows the generalization of domain features. We guess that the essence of the increase of confidence is to encourage the model closing to the target domain.
>
> ---
>
> **Question 2: Sensitivity to validation bilingual set in DCDPG**
>
> In Section 5.2, Table 3, Table 4, and Appendix C, we have already found and stated that the bilingual data do not always bring gains to CDPG. We also stated that CDPG is sensitive to changes in the top_p parameter, which affects its quality. In this case, Dynamic CDPG, as an auxiliary method, aims to optimize the hyper-parameter automatically using a small amount of bilingual data (validation set). Based on our design, if all optimizations are rejected in the validation set, DCDPG would fine-tune the model in the case where top_p = 1, which avoids heavy updating in the pre-trained models to ensure the bottom-bound of CDPG. Moreover, as indicated by the title, our main focus is CDPG, while Dynamic CDPG is presented as a variant and an attempt to automate optimal parameter settings.
>
> ---
>
> If you have any further concerns or feel that certain points were not addressed adequately, please don’t hesitate to let us know. We will respond sincerely.

---

> > ### Author Response · Authors · 2024-12-02
> >
> > Dear Reviewer VUPi,
> >
> > Thank you for your time and effort in reviewing our manuscript and engaging in the discussion process. As the discussion period nears its conclusion, we hope that our responses and revisions have effectively addressed your concerns.
> >
> > If there are any remaining questions or concerns, please let us know in detail at your earliest convenience. We will do our best to address them promptly before the discussion phase ends.
> >
> > If your concerns have been resolved, we kindly ask you to consider positively revising your evaluation to reflect the improvements in our work.
> >
> > Thank you again for your valuable feedback and dedication.
> >
> > Best regards,
> >
> > Anonymous Authors

---

> ### Author Response · Authors · 2024-12-04
> **Thank you for participating in the author-reviewer discussion phase**
>
> Dear Reviewer VUPi,
>
> We understand that you may be busy, and we greatly appreciate the time and effort you have already dedicated to this review process. We believe that your lack of response to our rebuttals is not due to irresponsibly abandoning your role as a reviewer, but rather because the concerns regarding this paper have been resolved. Sorry to bother you, but if the concerns have indeed been addressed, we kindly request you to **update your scores** for **Soundness**, **Presentation**, **Contribution**, and **Overall Rating** to reflect the resolution of all issues, as is typically expected of reviewers following the discussion period.
>
> Best regards,
>
> Anonymous Authors

---

### Official Review · Reviewer_oE5u · 2024-11-08

**Soundness:** 3
**Presentation:** 2
**Contribution:** 3
**Rating:** 6
**Confidence:** 3

**Summary:**

The paper introduces a method to perform domain translation for neural machine translation (NMT) by utilizing only monolingual domain-specific data. The authors employ energy-based models (EBMs) combined with Conditional Distributional Policy Gradients (CDPG) to perform domain adaptation without relying on large-scale parallel domain-specific data, which is often challenging to collect. The paper further proposes DYNAMIC CDPG to improve upon traditional CDPG by dynamically adjusting parameters using bilingual validation data, aiming to achieve optimal results without catastrophic forgetting. Experiments are conducted across several translation directions and domain adaptation scenarios.

**Strengths:**

Innovative Approach: The use of energy-based models combined with CDPG for domain adaptation with only monolingual data represents a novel approach that addresses a key limitation in domain-specific NMT.
Effective Results: The experimental results show that DYNAMIC CDPG performs well compared to fine-tuning and LORA-based methods, with improvements in key evaluation metrics such as BLEU and NIST scores.
Reduction in Catastrophic Forgetting: The proposed approach successfully mitigates catastrophic forgetting, which is a common issue in domain adaptation tasks, thus preserving the pre-trained model’s knowledge while adapting to a new domain.

**Weaknesses:**

Limited Scope of Evaluation: The experiments are conducted on a relatively small set of translation directions and domains. Expanding the evaluation to include additional languages and domains would provide stronger evidence for the generalizability of the approach.
Lack of Robust Comparison: The paper primarily focuses on comparisons with pre-trained, fine-tuned, and LORA baselines. It would be beneficial to include comparisons with other strong domain adaptation techniques such as back-translation and adversarial domain adaptation, which could provide a more holistic understanding of the strengths and weaknesses of the proposed methods.
Complexity of Methodology: The proposed methodology is relatively complex, and while the theoretical explanations are well-written, it may be challenging for readers without a background in reinforcement learning or energy-based modeling to fully grasp. Adding intuitive explanations or visual aids would enhance accessibility.

**Questions:**

Hyperparameter Analysis: While Table 2 provides insight into top-p values used for DYNAMIC CDPG, can the authors provide a deeper analysis on the sensitivity of other hyperparameters (e.g., learning rate, λ values for EBMs) and how they impact the final results? A sensitivity analysis or a hyperparameter optimization discussion could greatly strengthen the paper.

Computational Efficiency: How does the proposed approach compare in terms of computational cost to other domain adaptation methods, such as back-translation or adversarial domain adaptation? The paper mentions improvements in alignment quality, but a more explicit analysis of the computational trade-offs would be valuable.

Scaling to Larger Domains: Can the authors discuss the scalability of the proposed approach to much larger domain adaptation tasks? For example, would the methodology perform well with highly diverse target domains or significantly larger monolingual datasets?

Effectiveness of Monolingual Features: The paper leverages unigram frequency for domain adaptation, which is a relatively simple feature representation. Have the authors considered experimenting with more sophisticated feature representations, such as n-gram frequencies or embeddings? Would these improve the performance of CDPG and DYNAMIC CDPG for domain shifts?

Handling Noisy Monolingual Data: In real-world scenarios, monolingual domain data might be noisy. How robust is the proposed approach when dealing with noisy or imperfect monolingual data? Some discussion or experiments on the effects of noisy data would be insightful.

---

> ### Author Response · Authors · 2024-11-20
> **Rebuttal by Authors (1/2)**
>
> We greatly appreciate your insightful comments which have significantly contributed to improving our manuscript and enhancing its appeal to broader readers.
>
> ---
> **Weakness 1: Limited Scope of Evaluation**
>
> We conducted experiments across four language directions (en <-> de, en <-> zh) and four domains, resulting in a total of 16 tasks. Additionally, since the experimental data and settings we used align with those commonly employed in domain adaptation evaluations for machine translation, such as those mentioned in related work sections, we believe that our experiments are sufficiently comprehensive. Our experiments further support the general applicability of our approach in the context of domain adaptation studies in machine translation.
>
> ---
>
> **Weakness 2: Lack of Robust Comparison**
>
> From your comments, we have added the results of back-translation to Appendix F and Table 8, and mention it in Line 196 to strengthen our statements. Since methods for generating synthetic data, such as back-translation, are susceptible to noise, the performance of a model fine-tuned with clean data serves as an upper bound when the available sentences are the same. This trend is observed in the results of the additional experiments. On the other hand, our approach outperforms the fine-tuning results, concluding that it is superior to methods using monolingual data, such as back-translation.
> Moreover, based on your interesting Questions 3 and 5 about diversified and noisy data, we also added Appendix G and Table 9, which compare Fine-tuned and CDPG in a scenario of mixing two domains. The result shows that the vanilla fine-tuning methods are influenced by noisy data, in which the performance decreases with the increase of noisy data. However, our proposed method, CDPG, shows more robust performance under the same condition.
> Thank you for your feedback, which has allowed us to argue the effectiveness of our approach more robustly!
>
> ---
>
> **Weakness 3: Complexity of Methodology**
>
> Thank you for your feedback. From around line 86, we have tried to provide examples for a more intuitive explanation, but we will make an effort to make it even more intuitive.

---

> > ### Author Response · Authors · 2024-11-20
> > **Rebuttal by Authors (2/2)**
> >
> > **Question 1: Hyperparameter Analysis**
> >
> > Thank you. As part of a pilot study, we explored other parameters (e.g., learning rate, λ values for EBMs), which primarily influence the stability of training. Unless these parameters are set to extreme values, such as a learning rate that deliberately hinders training, the final results remain largely unaffected. In contrast, variations in the top_p value have a large impact on the final results. This is because scoring the generated sentences relies on diversity; generating a wider range of sentences leads to better scoring (as diversity for scoring is crucial in reinforcement learning). Given the direct influence of the trade-off between maintaining the quality of scored sentences and ensuring diversity on the final outcomes, we conducted an extensive investigation into the hyperparameter settings for top_p.
> >
> > ---
> >
> > **Question 2: Computational Efficiency**
> >
> > Indeed, compared to straightforward fine-tuning, our method is computationally more expensive because it requires generating sentences for scoring. However, even when compared to the most efficient fine-tuning approaches, the primary bottleneck lies in the sentence generation step, while the scoring part, involving vector production, has been engineered to a computationally negligible level. The slower generation speed is a common issue in methods like back-translation as well, so our method does not necessarily fall significantly behind when compared to other approaches. Additionally, it is worth noting that this study focuses on stress settings with limited domain data, where speed was not a critical factor. Nonetheless, your perspective is highly valuable and appreciated.
> >
> > ---
> >
> > **Question 3: Scaling to Larger Domains**
> >
> > This study utilizes the word frequency distribution of the target domain. As the amount of data increases, the distribution becomes closer to the true target domain, making the method more effective. However, the innovative aspect of this research lies in its ability to perform sufficiently well even in stress settings where only a small amount of data is used. Additionally, we conducted experiments on highly diverse target domains to examine whether the method works effectively even when domains are mixed. The discussion is mentioned in Weakness 2: Lack of Robust Comparison.
> >
> > ---
> >
> > **Question 4: Effectiveness of Monolingual Features**
> >
> > Your feedback is incredibly insightful. In our pilot study, we also tested other features, such as TF-IDF, and observed results similar to those using word distribution. For this study, we chose the simplest feature, word distribution, to demonstrate that using a large number of EBMs is effective for domain adaptation. Exploring other features are future work. We described it in the Future Works section.
> >
> > ---
> >
> > **Question 5: Handling Noisy Monolingual Data**
> >
> > Thank you very much. We believe this is a good point. To investigate how robust the method is against noise, we added a mix-domain setting (same as question 3). The discussion is mentioned in Weakness 2: Lack of Robust Comparison.
> >
> > ---
> >
> > The above are our responses to your concerns. We hope this addresses them to some extent, but if there are still any remaining issues, please let us know. We will sincerely address them to the best of our ability.

---

> > > ### Author Response · Authors · 2024-12-02
> > >
> > > Dear Reviewer oE5u,
> > >
> > > Thank you for your time and effort in reviewing our manuscript and engaging in the discussion process. As the discussion period nears its conclusion, we hope that our responses and revisions have effectively addressed your concerns.
> > >
> > > If there are any remaining questions or concerns, please let us know in detail at your earliest convenience. We will do our best to address them promptly before the discussion phase ends.
> > >
> > > If your concerns have been resolved, we kindly ask you to consider positively revising your evaluation to reflect the improvements in our work.
> > >
> > > Thank you again for your valuable feedback and dedication.
> > >
> > > Best regards,
> > >
> > > Anonymous Authors

---

> ### Author Response · Authors · 2024-12-04
> **Thank you for participating in the author-reviewer discussion phase**
>
> Dear Reviewer oE5u,
>
> We understand that you may be busy, and we greatly appreciate the time and effort you have already dedicated to this review process. We believe that your lack of response to our rebuttals is not due to irresponsibly abandoning your role as a reviewer, but rather because the concerns regarding this paper have been resolved. Sorry to bother you, but if the concerns have indeed been addressed, we kindly request you to **update your scores** for **Soundness**, **Presentation**, **Contribution**, and **Overall Rating** to reflect the resolution of all issues, as is typically expected of reviewers following the discussion period.
>
> Best regards,
>
> Anonymous Authors

---

### Author Response · Authors · 2024-11-27
**Kind Reminder**

Dear Reviewers,

As the deadline for discussion nears, we wish to reaffirm our dedication to addressing any unresolved concerns regarding our submission. We have thoroughly addressed all the latest concerns, and we believe these efforts have significantly strengthened our manuscript. **If our responses have resolved your concerns, we would greatly appreciate an updated evaluation.**

We recognize and appreciate the significant time commitment the review process requires, and we highly value your feedback. If there are any additional recommendations for enhancing our submission, we would be happy to take them into consideration.

Thank you for your time and thoughtful consideration,

Anonymous Authors

---

### Author Response · Authors · 2024-12-02
**Rebuttal Summary and Final Follow-up**

Dear Reviewers and ACs,

As the discussion period approaches its conclusion, we would like to provide a reminder and summarize our responses to the feedback received so far.

We sincerely thank all the reviewers and ACs for your diligent efforts and high-quality reviews. If you have any additional questions or require further clarification, please feel free to let us know. Your insights are highly valued.

---

We are delighted to note that reviewers have highlighted the following strengths in our paper:
- Innovative, novel, and extensible approach: Recognized as innovative (`oE5u`), novel (`VUPi`), extensible (`UmaA`), and a solid application (`UmaA`), with a well-justified and clearly described method (`n3SL`).
- Effective and extensive evaluation results: Highlighted for the robustness and thoroughness of the evaluation (`oE5u`, `VUPi`, `UmaA`, `n3SL`).
- Successful mitigation of catastrophic forgetting: Demonstrates effective domain adaptation while addressing catastrophic forgetting (Reviewers `oE5u`, `VUPi`).
- Performance improvement: Shows good performance compared to baselines (`oE5u`, `VUPi`, `UmaA`, `n3SL`).

In response to your valuable suggestions, we have conducted additional experiments and made several key modifications to the manuscript. For your convenience, we have highlighted these changes in green or red text in the revised manuscript:
- Back-translation results: Added in Appendix F and Table 8 (Addresses `oE5u` W2,` n3SL` W4, Q4).
- Noise and cross-domain settings: Added in Appendix G and Table 9 (Addresses `oE5u` W2, Q3, Q5, `VUPi` W1, n3SL W3).
- COMET Score: Included in footnote 12 of Section 4.3 and Appendix H (Addresses `VUPi` W2, `UmaA` Q2).
- Domain feature generalization discussion: Expanded in Appendix I (Addresses `VUPi` Q1, `n3SL` W2).
- Task definition and motivation for catastrophic forgetting: Elaborated in Section 1 Introduction (Addresses `UmaA` W2, `n3SL` W1, W4, Q2).
- General domain results: Added in Table 6 for a more robust evaluation of catastrophic forgetting (Addresses `UmaA` W2, `n3SL` W1, Q2, Q5).

For other individual suggestions and concerns, we have provided detailed responses for each reviewer in the comments section of the discussion.

Thanks to the insightful feedback from reviewers, we believe the revised manuscript has been significantly improved, making it a valuable and accessible contribution for a broad audience.

---

We deeply understand that the reviewing process is a volunteer effort, and we sincerely appreciate the time and effort you have devoted to providing feedback. This paper represents not only solid, insightful, and novel work but also a meaningful contribution to the research community. Your suggestions have helped us address its weaknesses and further strengthen its contributions.

**If your concerns have been addressed, we kindly request that you consider raising your score**. Should you have any remaining concerns, please do not hesitate to let us know. We are committed to addressing them sincerely and thoroughly within the remaining time. However, if we do not receive any further response, we will consider all concerns resolved. Thank you for your understanding and for engaging in a constructive discussion as the review process concludes.

We look forward to your reply.

Best regards,

Anonymous Authors

---

### Author Response · Authors · 2024-12-04
**Thank you for the Discussion Phase and Rebuttal Summary (1/2)**

Dear AC and reviewers,

We thank you for the time and effort you have dedicated to reviewing our work, which has been incredibly helpful in improving its quality. While we understand that reviewers may have been too busy during the rebuttal period, resulting in limited discussion, **we encourage further discussion during the subsequent AC-reviewer discussion phase** to confirm whether our rebuttal has adequately addressed the reviewers’ questions and concerns.

Additionally, we have sent several reminders and reached out to the reviewers multiple times regarding additional concerns before the deadline but did not receive any responses. Besides our active participation, **reviewers raised no further concerns during the discussion period**. Therefore, we believe that **all concerns have been resolved and that the reviewers are satisfied with our arguments**.

For your convenience, **to help the AC and reviewers more easily grasp the key points of the entire rebuttal, we provide a summary here**, hoping everyone can have a better understanding.

---

## **Summary of our work**

This paper proposes a novel domain adaptation method for neural machine translation (NMT) that leverages only small amounts of monolingual domain-specific data, addressing the challenges of obtaining parallel data. The approach employs energy-based models (EBMs) and Conditional Distributional Policy Gradients (CDPG) to approximate domain-specific features, represented as unigram distributions.  The experiments demonstrate that fine-tuning with a large number of EBMs achieves robust domain adaptation while avoiding catastrophic forgetting, with notable improvements in domain-specific word learning across various translation directions and scenarios.

---

## **Reviewers' positive comments**

- **Innovative, novel, and extensible approach**: Recognized as innovative (`oE5u`), novel (`VUPi`), extensible (`UmaA`), and a solid application (`UmaA`), with a well-justified and clearly described method (`n3SL`).
- **Effective and extensive evaluation results**: Highlighted for the robustness and thoroughness of the evaluation (`oE5u`, `VUPi`, `UmaA`, `n3SL`).
- **Successful mitigation of catastrophic forgetting**: Demonstrates effective domain adaptation while addressing catastrophic forgetting (Reviewers `oE5u`, `VUPi`).
- **Performance improvement**: Shows good performance compared to baselines (`oE5u`, `VUPi`, `UmaA`, `n3SL`).

---

> ### Author Response · Authors · 2024-12-04
> **Thank you for the Discussion Phase and Rebuttal Summary (2/2)**
>
> ## **Summary of Our Responses**
>
> We highlight key points of discussion and revisions made to address the reviewers’ concerns. Detailed responses to individual reviewers can be found in their respective response sections.
>
> We have addressed all raised concerns, and no additional concerns were raised by reviewers after our responses.
>
> ---
>
> ### **Reviewer oE5u**
>
> The reviewer’s primary concern was the lack of robust comparisons. In response, we added discussions on back-translation in **Appendix F and Table 8** as a new baseline and included experiments on mixed-domain scenarios in **Appendix G and Table 9**. These additions demonstrate that our method is robust and applicable to diverse domain adaptation settings.
>
> ---
>
> ### **Reviewer VUPi**
>
> The reviewer expressed concerns about robustness to noise, scalability, and mixed-domain extensions. To address this, we included mixed-domain results in **Appendix G and Table 9**, which illustrate the robustness, scalability, and extensibility of our method.
>
> Additionally, we introduced **COMET-based evaluations** in **footnote 12 of Section 4.3 and Appendix H**, which aligned with our previously reported results, further confirming the validity of our findings.
>
> ---
>
> ### **Reviewer UmaA**
>
> The reviewer participated in one round of discussion and raised two primary concerns: novelty and the definition of catastrophic forgetting.
>
> •	**Novelty**: We emphasized in the manuscript that this work is the first to successfully scale EBMs. Previous studies were limited to small-scale, synthetic experiments, but our method applies to challenging, real-world tasks in **unsupervised domain adaptation**. We demonstrated superior performance even under weak supervision settings like **Dynamic CDPG**, and our contributions extend beyond incremental improvements. By enabling scaling, we achieved a significant breakthrough in unsupervised domain adaptation.
>
> •	**Catastrophic Forgetting**: To address this, we added generic domain results in **Section 6.1 and Table 5** to show the stability of our method. We also clarified the definition of catastrophic forgetting in the **Introduction** (highlighted in green), showing that our method is effective both in reinforcement learning tasks and in downstream tasks.
>
> ---
>
> ### **Reviewer n3SL**
>
> The reviewer participated in one round of discussion and raised concerns about definitions (**Weakness 1**), the validity of en-de results (**Weakness 2**), and experimental suggestions (**Weakness 4**). (Weakness 3 was already addressed during the first discussion.)
>
> •	**Definitions (Weakness 1)**: We clarified the definitions in the **Introduction** (highlighted in green).
>
> •	**Validity of Results (Weakness 2)**: We supported our results with literature and added case studies in **Appendix I**, demonstrating their validity.
>
> •	**Experimental Suggestions (Weakness 4)**: We clarified that the suggested experiments had already been conducted and included in our results. Most concerns were definitional, and we addressed them thoroughly with additional manuscript revisions.
>
> ---
>
> ### **Conclusion of our discussions  and responses**
>
> Despite multiple reminders and more than a week since the final rebuttal, **no additional concerns were raised by any reviewer**. We take this as an indication that our responses have fully addressed all concerns and that our arguments have been accepted. If there were further concerns, it would have been the reviewers’ responsibility to engage in the discussion. The lack of additional comments strongly suggests that all concerns have been resolved and that our rebuttal has been accepted as satisfactory.
>
> We are also pleased to note that **all reviewers recognized the validity of our experiments and the effectiveness of our proposed method**. Their comments and suggestions allowed us to further refine our manuscript, making it clearer and more accessible to a broader audience.
>
> This concludes our summary. We hope for a constructive AC-reviewer discussion phase and a fair evaluation free from undue biases related to low scores or confidence.
>
> ---
>
> Thank you for your consideration.
>
> Best regards,
>
> Anonymous Authors

---

### Meta-Review · Area_Chair_FeSi · 2024-12-20

**Metareview:**

The paper presents an innovative method for domain translation in NMTusing energy-based models (EBMs) combined with Conditional Distributional Policy Gradients (CDPG), focusing on utilizing monolingual domain-specific data rather than parallel data. It also introduces a dynamic variant, DYNAMIC CDPG, which utilizes bilingual validation data for parameter adjustments to optimize results without catastrophic forgetting. The paper conducts experiments across multiple translation directions and domain adaptation scenarios.

While the authors argue the comprehensiveness of their experiments, expanding to more languages and domains could fortify the evidence of generalizability. Reviewers suggested incorporating a broader range of competitive baselines, such as back-translation and adversarial domain adaptation, for more robust comparisons. Some reviewers found the methodology complex, suggesting that more intuitive explanations or visual aids could be added for accessibility. Adding pseudocode or clearer implementation details would strengthen the paper. Also, emphasizing differences from existing works (e.g., Korbak et al., 2022) will help underline the novelty.

**Additional Comments On Reviewer Discussion:**

During the rebuttal, a detailed dialogue occurred regarding the scope of evaluation, comparisons to other domain adaptation methods, the nuances of catastrophic forgetting, and clarifications on various assumptions. The authors provided additional experiments, justifications, and manuscript improvements, while some reviewers remained skeptical about the alignment of claims with observed evidence, particularly concerning catastrophic forgetting and comparison baselines.

In the AE-Reviewer discussion, Reviewer UmaA still doesn't feel like this work contributes enough beyond applying the CDPG work to their own problem. Reviewer n3SL is also still leaning towards rejection for several reasons: 1) Limited applicability in real-world scenarios; 2) Evaluation on EN<->DE hinging on a "manually refined" test set; 3) Much of the motivation around catastrophic forgetting and out-of-domain performance is from citations of relatively old papers (2017-2021).

Ultimately, we decide to reject the submission.

---

### Decision · Program_Chairs · 2025-01-22

Reject